# The microbial metabolite desaminotyrosine protects against graft-versus-host disease via mTORC1 and STING-dependent intestinal regeneration

Changes in the intestinal microbiome and microbiota-derived metabolites predict clinical outcomes after allogeneic hematopoietic stem cell transplantation (allo-HSCT). Here, we report that desaminotyrosine (DAT), a product of bacterial flavonoid metabolism, correlates with improved overall survival and reduced relapse rates in patients receiving allo-HSCT. In preclinical mouse models, treatment with synthetic DAT prevents graft-versus-host disease by protecting the intestinal barrier and promoting intestinal regeneration and contributes to graft-vs.-leukemia responses. DAT´s beneficial effects on intestinal regeneration remain effective despite broad-spectrum antibiotics-induced dysbiosis, also when administered by fecal microbiota transfer with flavonoid-degrading *F. plautii*. Mechanistically, DAT promotes mTORC1-dependent activation and proliferation of intestinal stem cells, with concomitant engagement of the innate immune receptor STING required to mitigate metabolic stress and maintain an undifferentiated stem cell state independently of type-I interferon responses. Additionally, DAT can skew T cells towards an effector phenotype to modulate graft-versus-leukemia responses. Our data uncover DAT's dual, tissue- and immune-modulating properties and underscore its potential in precision microbiome-based therapies to improve tissue regeneration and minimize immune-mediated side effects.

Allogeneic hematopoietic stem cell transplantation (allo-HSCT) is a curative therapy for many otherwise incurable hematological cancers. Yet, its success is limited by the severe morbidity and mortality associated with graft-versus-host disease (GvHD), characterized by an allogeneic immune reaction of donor T cells directed against the recipient's organs, especially the skin, intestine, and liver[1]. However, donor T cells are critical in eliminating residual malignant cells (graft-versus-leukemia (GvL)). Cancer relapse is the primary cause of death in patients receiving allo-HSCT, followed by GvHD[2]. Immunosuppressants, corticosteroids, and JAK 1/2 inhibitors are used to treat acute GvHD[3], but these drugs dampen critical GvL responses, increasing the risk of relapse[4]. Developing treatments that prevent deleterious GvHD while retaining GvL responses remains an unmet need.

Numerous studies have shown that changes in the intestinal microbiome, including the loss of diversity and expansion of disease-related taxa, are common in patients receiving allo-HSCT[5–8]. These GvHD-related microbial signatures have been attributed to broad-spectrum antibiotics and radio-/chemotherapy-based conditioning regimes[9,10] but may also be influenced by malnutrition or

✉e-mail: erik.orberg@ukr.de; hendrik.poeck@ukr.de

medication[11,12]. Microbiome diversity and composition have been associated with clinical outcomes after allo-HSCT, including overall survival and incidence of GvHD[13–16]. In contrast, little is known about whether and how the microbiome modulates GvL, and few studies have shown an association between microbiota and relapse[17–19]. How functional microbiome alterations impact the outcomes of patients receiving allo-HSCT is still not fully understood.

Microbiota-derived metabolites can exert tissue and immune-modulatory function and modulate outcomes following allo-HSCT[19–22]. Our group recently established an "Immuno-Modulatory Metabolite Risk Index (IMM-RI)" comprised of five metabolites (including the short/branched-chain fatty acids butyric, propionic and isovaleric acid, the tryptophan derivate indole-3-carboxaldehyde (ICA) and the flavonoid-metabolism byproduct desaminotyrosine (DAT)) that were predictive of overall survival and relapse in patients receiving allo-HSCT[19]. In mice, butyric acid and indoles (e.g., ICA) mitigate GvHD by improving mucosal barrier function, increasing intestinal regulatory T cells (Tregs), and inducing anti-inflammatory cytokines via aryl hydrocarbon receptor activation[23–31], respectively. In contrast, DAT enhances anti-viral immunity by amplifying type-I interferon (IFN-I) signaling and boosts immune-checkpoint blockade (ICB) efficacy in murine tumor models[32–35]. Yet, the biological relevance of DAT in the context of allo-HSCT is unclear.

Here, we describe a "dual function" of DAT which can improve tissue regeneration while achieving moderate antitumor responses and characterize its associated molecular mechanisms. DAT´s dual function is unique among other reported microbial metabolites in allo-HSCT, suggesting a potential broad applicability as a therapeutic agent to modulate efficacy and toxicity and as a clinical biomarker for predicting outcomes in patients receiving cellular immunotherapies.

## Results
### The microbial metabolite DAT modulates clinical and experimental GvHD and GvL
Utilizing targeted mass spectrometry (MS), we quantified fecal concentrations of DAT and ICA in 50 patients (patient characteristics in Table S1) before (day −7) and after allo-HSCT (obtained between days +7 to +21, Fig. 1A). Both metabolites declined significantly after allo-HSCT (Fig. 1B). Based on previously established thresholds (DAT = 0.005 µmol/g dry feces; ICA = 0.035 µmol/g dry feces)[19], we categorized patients into high and low DAT or ICA groups (Table S1). High concentrations of DAT and ICA correlated with improved two-year overall survival (Fig. 1C).

We then explored whether DAT treatment could protect mice receiving murine allogeneic bone marrow transplantation (allo-BMT) from experimental GvHD induced by major histocompatibility complex (MHC) mismatch (C57BL/6 donors transplanted into BALB/c recipients). Recipients received donor bone marrow (BM) and T cells, while a control group received only donor BM. DAT was administered daily via oral gavage from day −7 before allo-BMT until day 0, and ICA was administered as described before (day −7 until day +12 after allo-BMT[28]) (Fig. 1D). The T cell control group received oral gavage with the carrier only. Treatment with DAT or ICA significantly mitigated GvHD-induced weight loss (Fig. S1A) and prolonged overall survival compared to carrier-treated mice (Fig. 1E).

Given that the therapeutic effect of allo-HSCT relies on anti-tumor responses exerted by donor T cells, we further explored the impact of metabolites on GvL using the Ba/F3 FMS-like tyrosine kinase 3-internal tandem duplication-luciferase (Ba/F3 FLT3-ITD-Luc) acute lymphoid leukemia (ALL) model. BALB/c recipients were inoculated with syngeneic Ba/F3 leukemia cells and received donor BM. We transplanted allogeneic T cells (C57BL/6→BALB/c) or PBS 2 days post inoculation. As before, mice were treated with metabolites or carrier alone (Fig. 1F), and the ensuing Ba/F3 leukemia cell expansion was monitored via bioluminescence imaging (BLI, Figs. 1G, H, and S1B). As expected, we

observed a significantly reduced tumor burden and prolonged survival in mice that received allogeneic T cells than in those that received BM only (Fig. 1I). DAT treatment delayed tumor progression, resulting in moderately prolonged survival compared to mice transplanted with allogeneic T cells or BM only. Consistent with previous reports, ICA treatment did not negatively affect GvL responses[28] (Fig. 1H, I). We further confirmed the beneficial effect of DAT treatment in the FLT3-ITD/MLL-PTD AML model. As before, DAT treatment significantly reduced relapse rates compared to T cell-treated or BM only mice without inducing GvHD (Fig. S1C–E).

Given that DAT improved GvL responses in mice, we assessed the incidence of relapse in our allo-HSCT patient cohort stratified according to high and low DAT concentrations, adjusting for transplantation-related mortality (TRM) as a competing risk. High DAT concentrations were independently associated with significantly lower relapse rates in patients receiving allo-HSCT (Fig. 1J). These results indicate that DAT may exert dual roles in allo-HSCT, protecting from GvHD while contributing to GvL.

The intestinal epithelium, located at the host-microbiome interface, plays a crucial role in GvHD-related morbidity when damaged. Preserving the integrity of the intestinal barrier, such as through metabolites or IFN-I agonists, has proven effective in mitigating GvHD[25,28,36]. We therefore next focused on DAT's impact on the intestinal epithelium.

### DAT treatment promotes intestinal organoid growth and protects against toxicity in vitro
To model the impact of metabolite treatment on the intestinal barrier, we administered DAT to murine small intestinal crypt-derived (SI) organoids (Fig. 2A). As before, we used ICA as a positive control for a metabolite known to mitigate GvHD in an IFN-I-dependent manner. Metabolite stimulations were performed with the median stool metabolite concentration assessed in patients at day −7 before allo-HSCT (~50 µM ICA and 100 µM DAT) (Fig. 1B).

DAT treatment of organoids significantly increased organoid size (Fig. 2B) and the abundance of Lgr5-GFP^high intestinal stem cells (ISCs) (Figs. 2C and S2A) as a surrogate for increased proliferation. Consistently, DAT treatment enhanced organoid expansion, indicated by increased numbers of organoids that were established after passage compared to untreated organoids (Fig. 2D). In contrast, ICA treatment exhibited no effect on the size, ISC abundance or organoid numbers (Fig. 2B–D).

To examine whether DAT could protect the intestinal epithelium, we challenged murine organoids with different toxic regimens relevant in patients receiving allo-HSCT in the presence or absence of metabolites.

T-cell toxicity is mediated by inflammatory cytokines such as IFNγ[37]. To mimic the toxic effect of these soluble mediators on the epithelium, we co-cultured C57BL/6 SI organoids with CD3/28-bead activated allogeneic CD3+ T cells (magnetic cell (MACS)-separated from BALB/c spleens) suspended in the surrounding media but without direct contact between them. As a result, the expansion of organoids after passaging was reduced compared to control organoids cultured without T cells. Treatment with DAT rescued organoid regeneration, while ICA was less effective (Fig. 2E).

Donor CD8+ T cells can infiltrate the recipient intestinal epithelium and target epithelial cells in a direct, TCR-MHC-mediated allo-reaction. To model this interaction, we co-cultured intestinal organoids with allogeneic intraepithelial T cells for 48 h in direct contact[38] and assessed cell death in organoids by propidium iodide staining (normalized for cell count using Hoechst staining, Fig. 2F, right). In this model, treatment with ICA, but not DAT, protected organoids from direct, T cell-mediated cytotoxicity (Fig. 2F, left). This may be explained in part by ICA-induced downregulation of MHC class II expression in organoids (Fig. S2B).

To mimic conditioning damage, we exposed organoids to irradiation alone or in presence of metabolites. We observed that

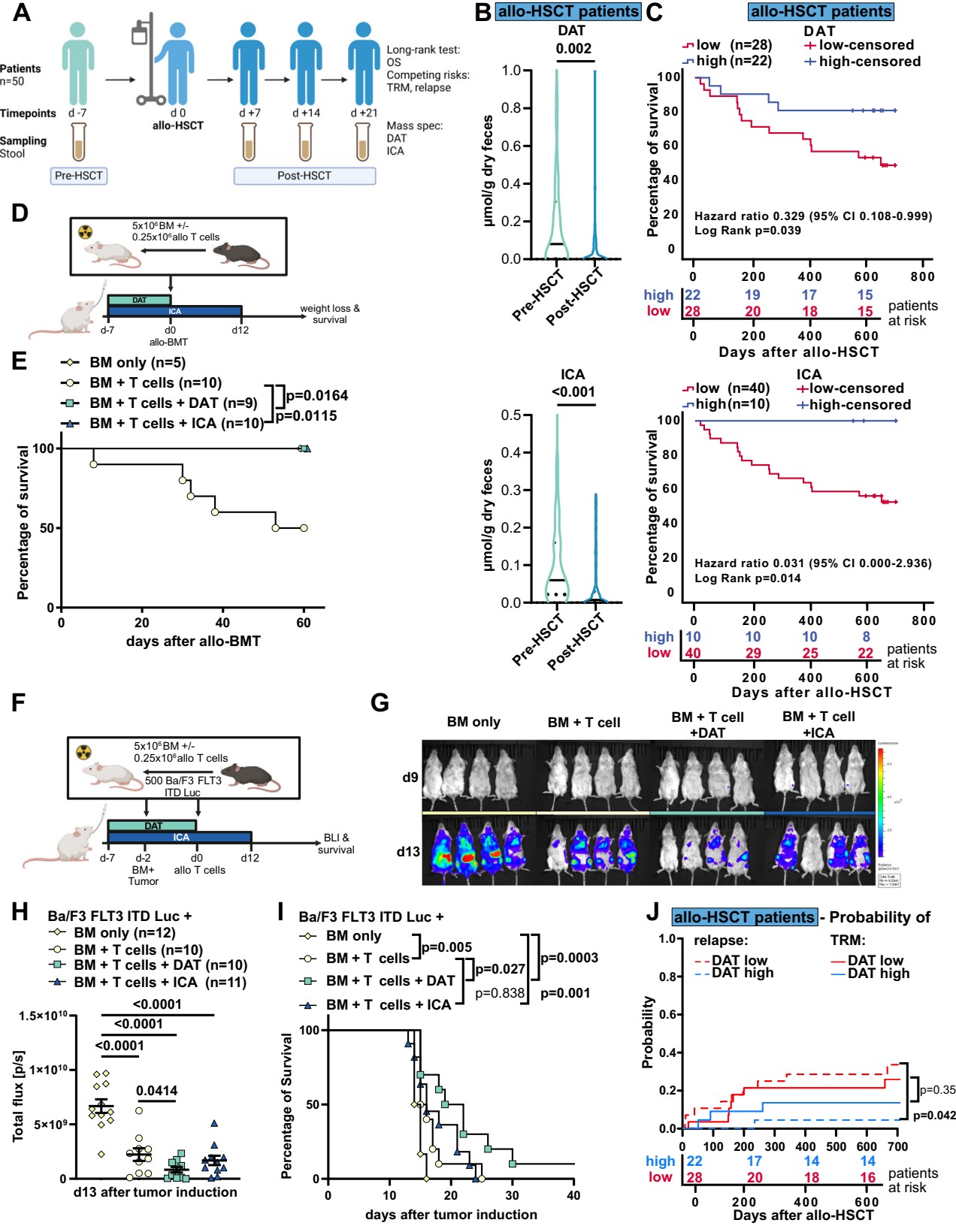

treatment with DAT but not ICA improved the number of viable organoids and organoid expansion following irradiation (Fig. S2C).

These findings suggest that DAT promotes regeneration of the intestinal epithelium in vitro and protects it from radiation and inflammatory cytokines without affecting the direct interaction between T cells and epithelial cells.

## DAT treatment protects the intestinal barrier and stem cell compartment from tissue damage in vivo

To investigate the impact of metabolites on the intestinal epithelium in vivo, we administered DAT and ICA to mice and assessed barrier damage and the regenerative capacity of the ISC compartment after allo-BMT (Fig. 2G).

**Fig. 1 | The microbial metabolite DAT modulates clinical and preclinical GvHD and GvL. A** Study overview. **B** Stool metabolite concentrations in patients receiving allo-HSCT before (day −7) and after (day +7 to +21) transplantation measured by targeted mass-spectrometry in 50 patients from Munich and Regensburg. Data is presented as violin plot with median as line and quartiles as dotted lines and analyzed by non-parametric paired *t*-test. **C** Kaplan Meier plots of patient's overall survival after allo-HSCT grouped by stool metabolite concentrations of DAT or ICA based on established cut-offs (DAT = 0.005 μmol/g dry feces; ICA = 0.035 μmol/g dry feces) and corresponding patients at risk table. Data was analyzed by Log-rank test. **D** Experimental design of the murine MHC major mismatch allo-BMT model. **E** Survival of mice after allo-BMT. Pooled data from 2 independent experiments. Analyzed by Log-rank test. **F** Experimental design of the GvL model in (**G–I**).

**G** Representative BLI images and **H** quantification of BLI signal on d13 from two independent experiments. Further time points are shown in Fig. S1B. *P*-values were calculated using two-way ANOVA with Dunnett's correction for multiple comparisons or unpaired *t*-test. **I** Kaplan Meier plot of survival. Data was analyzed by Log-rank test from 2 independent experiments. **J** Cumulative incidence of relapse and TRM after 2 years after day +21 in a competing risk analysis, stratified according to DAT high and DAT low. Cumulative incidence functions between groups were tested for equality using Gray's test. If not depicted otherwise, all data are shown as mean ± SEM. **A** Created in BioRender. Göttert, S. (2025) https://BioRender.com/lgwl8o0. **D** Created in BioRender. Göttert, S. (2025) https://BioRender.com/cxfhu92. **F** Created in BioRender. Göttert, S. (2025) https://BioRender.com/c784lxh.

Mice received total-body irradiation (TBI) and the influx of CD11b+Ly6G+ granulocytes to the intestinal lamina propria was assessed at day +3 as a surrogate of intestinal barrier damage[39]. Metabolite treatment reduced granulocyte influx compared to carrier-treated mice (Fig. 2H).

To assess the regenerative capacity of the ISC compartment, we developed an organoid-based assay in which intestinal crypts were isolated from the small intestine on day +7 after allo-BMT, and a defined crypt number was seeded into organoid cultures. After 72 h, organoid numbers were counted. A higher number of organoids established is suggestive of more intact ISCs. As expected, transplantation of allo-BM and T cells resulted in reduced SI organoid regeneration compared to allo-BM alone (Fig. 2I). However, treatment with DAT or ICA in mice that received BM and allogeneic T cells restored organoid regeneration to levels comparable with BM alone (Fig. 2I). We confirmed this protective effect in organoids derived from large intestinal (LI) crypts (Fig. S2E). Similar results for barrier integrity and SI organoid recovery were obtained when intestinal damage was induced by TBI and syngeneic BMT and the anthracycline doxorubicin (Fig. S2F, G).

Histological analysis of the large intestine revealed a reduction in GvHD histopathological scoring (Fig. 2J) and T cell infiltration (Fig. 2K) as well as preservation of the mucus layer (Fig. 2L) in DAT or ICA-treated mice compared to controls. Systemically, metabolite treatment significantly reduced serum IFNγ concentrations at day +7 after allo-BMT. In addition, DAT treatment resulted in a trend towards reduced TNFα and elevated IL-10 concentrations in serum (Fig. S2H).

In summary, these findings suggest that DAT provides in vivo protection by safeguarding the intestinal epithelium from various forms of damage and supporting epithelial regeneration, ultimately reducing GvHD-related pathology.

## DAT treatment remains effective despite the use of broad-spectrum antibiotics

The interplay of the bacteriome, fungome, and virome correlates with metabolite production, while conversely, metabolite treatment itself may shape intestinal microbial communities[19,32]. Therefore, we assessed whether metabolite treatment affected the murine intestinal microbiome. Daily metabolite treatment in mice via oral gavage for seven days did not significantly alter the family-level taxonomic composition compared to carrier treatment (Fig. 3A). Similarly, neither alpha diversity (Fig. S3A, B) nor beta diversity (Fig. S3C) differed significantly between metabolite- or control-treated mice.

To investigate whether the protective effect of DAT required an intact microbiome, we treated germ-free (GF) mice with bacterial metabolites, performed allo-BMT, and assessed organoid regeneration as before (Fig. 3B). The efficacy of metabolites was strongly reduced in GF mice: DAT treatment resulted in a non-significant trend towards improved organoid recovery, while ICA had no effect (Fig. 3B).

Since epithelial and immune functions are altered in GF mice[40], we assessed whether colonization with microbiota may be required for them to respond to metabolite treatment. Therefore, we isolated intestinal crypts from GF mice that received fecal microbiota transplantation (FMT) from specific-pathogen-free (SPF) donor mice seven days earlier or a mock-FMT and stimulated isolated crypts ex vivo with metabolites directly after isolation. SI crypts were responsive to DAT treatment, whether they were isolated from FMT-recipients or GF mice (Fig. 3C). This suggested that the poor efficacy of metabolites in GF mice may be due to an inherent lack of microbial signals (e.g., microbe-associated molecular patterns) or impaired intestinal development.

To overcome this limitation, we performed gut decontamination in SPF mice using an antibiotic cocktail consisting of ampicillin and enrofloxacin administered via drinking water beginning seven days before initiating metabolite treatment (Fig. S3D). This resulted in the complete suppression of culturable bacteria and no amplification of 16S rRNA bacterial reads (Fig. S3E, F)[41]. In this setting, the protective effect of DAT in mice receiving allo-BMT was restored (Fig. 3D).

Since gut decontamination in patients receiving allo-HSCT was ineffective in reducing mortality, it is no longer the standard of care[42]. However, the use of broad-spectrum antibiotics remains widespread in these patients and is associated with dysbiosis and GvHD-related microbial signatures[5–8]. To test whether metabolite treatment remains effective during therapy with broad-spectrum antibiotics, we administered meropenem to mice for seven days before initiating metabolite treatment and performing allo-BMT (Fig. S3G). Administration of meropenem aggravates colonic GvHD[43]. Consistently, meropenem significantly hampered organoid regeneration, reducing LI crypt-derived intestinal organoids compared to mice that did not receive antibiotics. However, metabolite treatment with DAT or ICA restored organoid recovery, compensating for antibiotic-induced dysbiosis (Fig. 3E). To further validate the protective effect of DAT under dysbiotic conditions, we treated mice as previously described and applied our major mismatch GvHD model (Fig. S3H). DAT treatment significantly enhanced survival following meropenem-induced dysbiosis compared to meropenem treatment alone (Fig. 3F).

FMT is approved in antibiotics-refractory *C. diff.* colitis and in clinical trials for gastro-intestinal GvHD[44]. Although the exact mechanisms are not fully understood, mitigating the adverse effects of broad-spectrum antibiotics and restoring microbial diversity may be important. Given that the exogenous administration of chemically synthesized DAT could enhance intestinal regeneration despite dysbiosis, we investigated whether a "precision" FMT using a flavonoid-degrading, DAT-producing bacterial probiotic[33,45] as a physiological source of DAT might be equally effective. After exposure to ampicillin and enrofloxacin, mice received suspensions of *Flavonifractor plautii* (*F. plautii*) strains isolated from human (DSM 6749) or murine (DSM 26117) feces or control FMT with cecal content from SPF mice followed by allo-BMT (Fig. S3I). Human-isolated *F. plautii* administration significantly improved organoid regeneration on day seven after allo-BMT compared to control FMT (Fig. 3G).

These findings suggest that metabolite-based treatments remain effective even after exposure to broad-spectrum antibiotics and that intestinal regeneration can be enhanced through synthetic compounds or flavonoid-degrading bacteria.

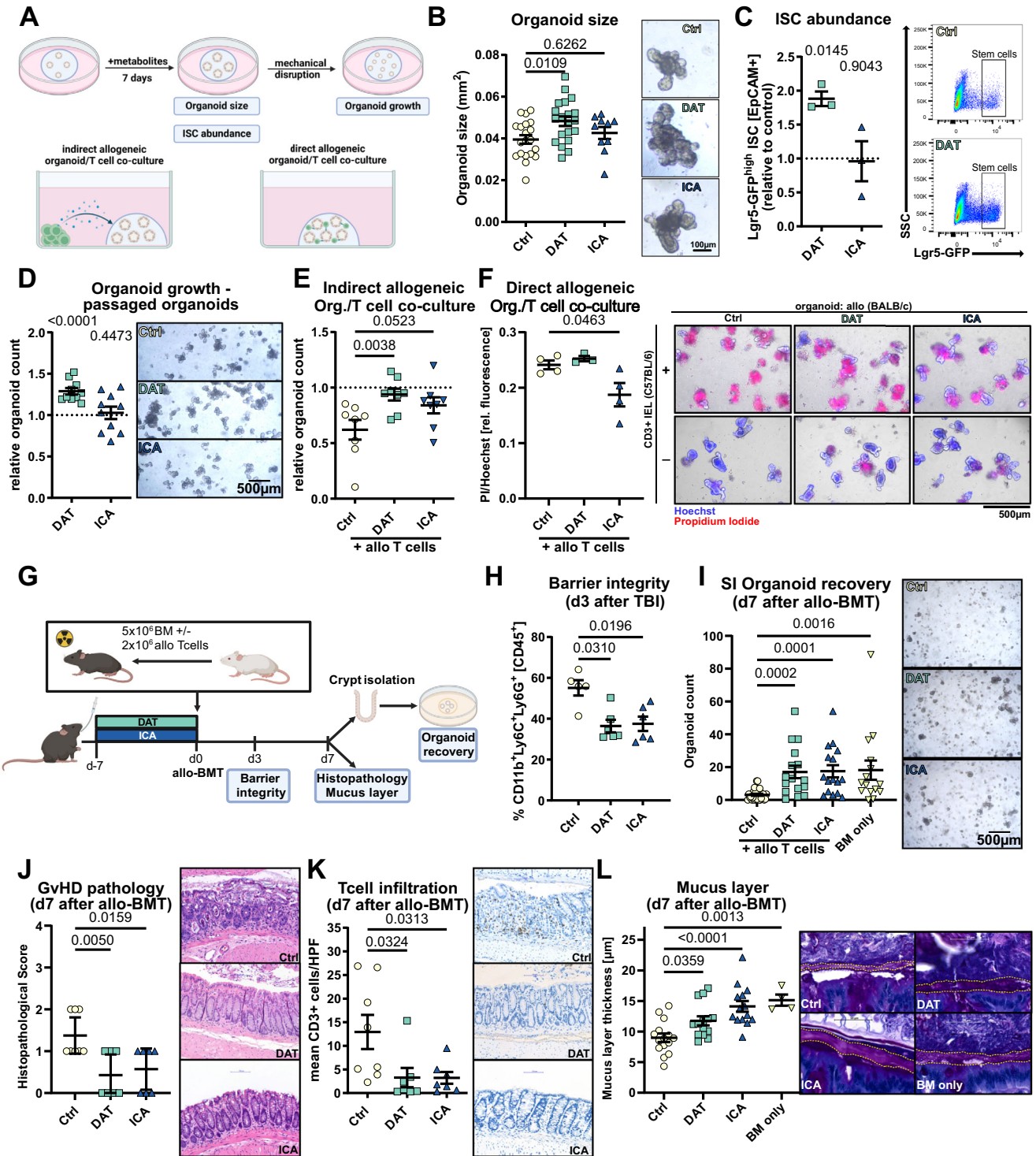

## DAT-induced organoid regeneration requires intact STING but not IFN-I signaling

Given the known link between DAT and IFN-I responses[28,32,33], we next investigated whether intact IFN-I signaling or IFN-I-inducing pathways were required for DAT's protective effect on the intestinal epithelium. To this end, we utilized intestinal organoids deficient for the IFN-I receptor (IFNaR[−/−]), the stimulator of interferon genes (STING Gold-enticket, STING[GT/GT]), or the mitochondrial antiviral-signaling protein (MAVS[−/−]).

The improved growth of DAT-treated SI organoids was abrogated in the absence of STING signaling (Fig. 4A, left). In contrast, DAT's effect did not depend on intact IFNaR (Fig. 4A, right) nor MAVS

signaling (Fig. S4A). Similarly, in the absence of functional STING, DAT treatment did not significantly increase organoid size (Fig. 4B). Like in wild-type (WT) organoids, ICA treatment did not affect IFNaR[−/−], STING[GT/GT], or MAVS[−/−] organoid growth.

To examine whether STING signaling was required for DAT's protective effects, we first co-cultured STING-deficient organoids with WT allogeneic T cells in the surrounding media (without direct contact between organoid and T cells, as in Fig. 2E). The protective effect of DAT against inflammatory cytokines was abrogated in the absence of epithelial STING signaling (Fig. 4C). We further investigated the impact of STING deficiency on DAT's protective effect against radiotoxicity as before (Fig. S2C). As expected, in STING KO organoids, DAT no longer

**Fig. 2 | DAT promotes growth and damage protection in murine organoids and in vivo damage models. A** Experimental overview. **B–D** Murine SI organoids were stimulated with 100 µM DAT, 50 µM ICA or carrier. **B** Representative images and quantification of organoid size after 6 days of metabolite treatment. Data pooled from 10 (ICA) or 20 (Ctrl and DAT) independent experiments. **C** ISC abundance assessed by flow cytometry (Fig. S2A) after 6 days of metabolite treatment of organoids derived from Lgr5-GFP reporter mice. Representative FACS plots and abundance of Lgr5-GFP^high ISCs relative to untreated organoids from 3 independent experiments are shown. **D** Number of established organoids after passaging relative to control (indicated by the dotted line). Data was pooled from 10 independent experiments. **E** Murine SI organoids were co-cultured with activated splenic T cells in the surrounding media for 4 days without direct interaction and their growth was assessed as described in (**D**). Data was pooled from 8 independent experiments. **F** SI organoids were brought in direct contact with allogeneic intraepithelial T cells. After 48 h, cell death was detected by propidium iodide fluorescence and normalized to Hoechst fluorescence. Representative images and PI/Hoechst quantification. Pooled data from 3 (DAT) or 4 (Ctrl and ICA) independent experiments.

**G** in vivo experimental design. **H** Infiltration of CD11b+Ly6G+Ly6C+ granulocytes of total live CD45+ cells in ileum was assessed by flow cytometry (Fig. S2D) on day 3 after 9 Gy TBI. Pooled data from 3 (n = 5 (Ctrl) or 6 (DAT and ICA)) independent experiments. **I** Count of recovered small intestinal organoids on day 7 after allo-BMT and representative pictures. Pooled data from 6 experiments (n = 29 (Ctrl), 15 (DAT and ICA) and 14 (BM only)). **J** Histopathology score and **K** number of infiltrating T cells in the colon of mice treated as in (Fig. 2G). Pooled data from 3 independent experiments (n = 8 (Ctrl) or 7 (DAT and ICA)). Scale bar indicates 100 µm. **L** Quantification and representative pictures of the large intestinal mucus layer thickness. Pooled data from 3 independent experiments (n = 15 (Ctrl), 14 (DAT and ICA) or 4 (BM only)). Scale bar indicates 50 µm. *P*-values were calculated by ordinary one-way ANOVA with Dunnett's correction for multiple comparisons (**B, E, F, J, K, L**), Kruskal–Wallis test with Dunn's correction for multiple comparisons (**H, I**) or one sample *t*-test (**C, D**). All data are shown as mean ± SEM. **A** Created in BioRender. Göttert, S. (2025) https://BioRender.com/era0sp5. **G** Created in BioRender. Göttert, S. (2025) https://BioRender.com/wzxx6td.

---

increased the number of viable organoids or the expansion following irradiation (Fig. S4B).

To confirm these findings in vivo, we performed metabolite treatment in IFNaR^−/− or STING^GT/GT mice (C57BL/6 background), followed by allo-BMT from WT BALB/c donors. We observed that DAT-induced organoid regeneration was abrogated in STING^GT/GT recipients but remained intact in recipient mice lacking IFNaR (Fig. 4D). To confirm that intestinal STING expression is necessary for the protective effect of DAT, we crossed STING^fl/fl and Vilin-Cre mice to generate a tissue-specific knockout in intestinal epithelial cells (STING^ΔIEC). Similar to STING^GT/GT mice, the protective effect of DAT was lost in these mice (Fig. S4C). In contrast to previous reports, the protective effect of ICA on the ISC compartment was not dependent on IFN-I and remained intact in both knockout models[28].

In summary, STING loss blocked DAT-driven organoid growth and protection from inflammation and radiation, showing that DAT's intestinal benefits rely on epithelial STING signaling, but not on IFN-I or MAVS pathways.

## DAT promotes the expansion of metabolically-active intestinal stem cells

To better understand DAT's effect on intestinal organoids and the role of STING in this context, we conducted single-cell RNA sequencing (scRNA-seq) of WT and STING-deficient organoids after treatment with DAT or control (Fig. 4E). We annotated cells via a combination of graph-based clustering, evaluation of marker signature expression, and automated cell type annotation based on a scRNA-seq reference dataset from the murine intestine[46] (see *Methods* for a detailed description of the approach). This enabled us to identify all major intestinal epithelial cell types (Fig. 4F).

Using this approach, cells with stem cell characteristics, namely ISCs and transit-amplifying cells (TA), resided on a continuum and were poorly separated. To resolve this continuum and enhance the precision of cell type annotation, we conducted trajectory analysis on the subset of cells initially classified as ISC or TA. Cells were ordered in pseudotime according to their expression of genes that differentiate ISC and TA populations in the reference dataset[46]. Based on trajectory states in pseudotime and their expression of stem cell markers, we categorized these stem-like cells into three groups: ISC, ISC II, and TA (Fig. 4G, detailed description of the approach and the branching of ISC II and TA on component 2 in the methods section). Here, ISC and TA correspond to the classical definition of ISCs and TA cells, while ISC IIs represent an intermediate population.

Next, we examined whether metabolite treatment altered the abundance of these cell types. In WT organoids, treatment with DAT led to an expansion of ISC-II. This effect was attenuated in STING-deficient organoids. In direct comparison, the abundance of ISC-II was reduced in DAT-treated STING-deficient (Goldenticket (GT) DAT)

versus DAT-treated WT organoids (WT DAT). Conversely, we observed an increased abundance of differentiated enterocytes in GT DAT vs WT DAT (Fig. 4H).

Considering that DAT strongly affected ISC-IIs, we focused on them for detailed analysis. In gene expression analysis and gene set enrichment analysis (GSEA), ISC-IIs showed high expression of the ISC markers Lgr5 and Olfm4 (Fig. S4D), accompanied by upregulation of gene sets associated with metabolic and proliferative pathways (Fig. S4E). To confirm an increased metabolic activity, we measured the media oxygen concentration in DAT-treated organoid cultures as a surrogate for oxygen consumption. As expected, DAT treatment increased the oxygen consumption of wild-type organoids, but not of STING-deficient organoids (Fig. S4F). When organoids were additionally challenged by irradiation, we observed a reduced oxygen consumption compared to control organoids. In irradiated organoids, DAT treatment attenuated the irradiation-induced decrease in oxygen consumption (Fig. S4G).

## STING regulates DAT-activated mTORC1 signaling and cellular stress in intestinal stem cells

According to our GSEA results, the transition from ISC to ISC-II, as well as from ISC-II to TA cells, was linked to an upregulation of genes associated with mTORC1 signaling in steady-state organoids. mTORC1 is a well-established regulator of cellular metabolism and proliferation[47,48] (ISC-II vs. ISC and TA vs. ISC-II, Fig. S4E). In the transition from ISC over ISC-II to TA cells we observed upregulation of gene sets associated with metabolic activity via GSEA, specifically oxidative phosphorylation (OXPHOS). OXPHOS is required for energy production, but also generates reactive oxygen species (ROS), damaging mitochondria and releasing mitochondrial DNA (mtDNA) in the process. Consequently, we also observed an upregulation of genes associated with cellular stress pathways, particularly related to ROS, and pathways for organelle maintenance and self-renewal through autophagy, specifically mitophagy (i.e., the selective degradation of mitochondria by autophagy, Fig. S4E) during the transition from ISC through ISC-II to TA.

Given that DAT promoted ISC-II abundance and genes associated with mTORC1 signaling are increased in ISC-II compared to ISC, we hypothesized that DAT may require mTORC1 to mediate its effects. To test this, we treated SI organoids with the mTOR inhibitor rapamycin. Rapamycin abrogated DAT-induced organoid growth (Fig. 5A), and inhibition of the mitogen-activated protein (MAP) kinase pathway, an upstream activator of mTORC1, with U0126, had the same effect. Consistently, rapamycin compromised the expansion of Lgr5^high ISCs in response to DAT (Fig. 5B).

Next, we used GSEA to assess the effect of DAT treatment compared to untreated control within the individual cell types annotated in

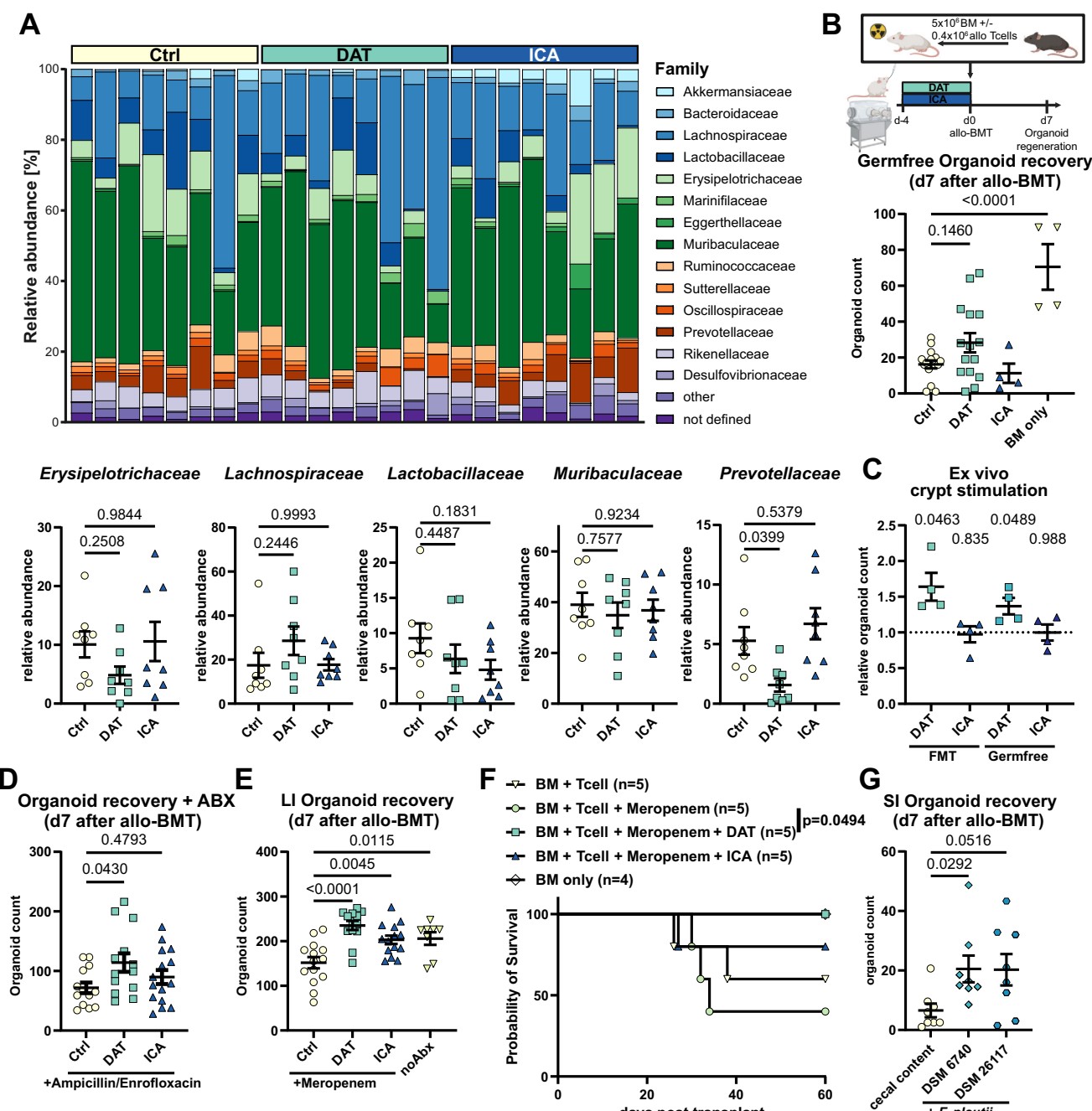

**Fig. 3 | DAT treatment is effective in SPF but not GF mice and remains effective after antibiotic-induced dysbiosis. A** Mice were treated with metabolites as described in Fig. 2G, stool samples ($n = 8$) were collected on d0 and analyzed by 16S rRNA sequencing. Relative abundance on family level and comparisons of the top 5 most abundant families is shown. **B** Experimental layout for treatment of germfree mice and organoid regeneration was assessed as described for Fig. 2I but Rock inhibitor was added to the culture medium. Pooled data from 4 experiments ($n = 15$ (Ctrl and DAT) or 4 (ICA and BM only)). **C** Small intestinal crypts were isolated from GF BALB/c mice or littermates reconstituted by FMT. Crypts were stimulated directly after initiation of culture and organoid growth was assessed as in Fig. 2D relative to its respective control. Pooled data from 4 independent experiments. **D** C57BL/6 mice were continuously treated with ampicillin/enrofloxacin added to their drinking water beginning one week before metabolite treatment. Allo-BMT and organoid recovery were performed as in Fig. 2I but in presence of Rock inhibitor. Pooled data from 3 independent experiments ($n = 13$ (Ctrl and DAT) or

15 (ICA)). **E** C57BL/6 mice were treated with meropenem per drinking water for 7 days for induction of dysbiosis before metabolite treatment, allo-BMT with $1.5 \times 10^6$ allogeneic T cells and large intestinal organoid regeneration was performed as outlined in Fig. S2E. Pooled data from 3 independent experiments ($n = 14$ (Ctrl and ICA), 13 (DAT) or 8 (noAbx)). **F** Balb/c mice were treated with meropenem and metabolites as in (**E**) and underwent allo-BMT. Kaplan–Meier plot of survival. Data was analyzed by Log-rank test. **G** Mice were treated with ampicillin/enrofloxacin for 5 days beginning on day -14. On day −7 and -5, mice received FMT with cecal content of SPF mice or human or murine *F. plautii* isolates (DSM 6740 or 26117, respectively) followed by allo-BMT and scoring of SI organoid regeneration. Pooled data from 3 independent experiments ($n = 7$). *P*-values were calculated with ordinary one-way ANOVA with Dunnett's correction for multiple comparisons (**A**, **B**, **D**, **E**), Kruskal–Wallis test with Dunn's correction for multiple comparisons (**G**) or one sample *t*-test (**C**). All data are shown as mean ± SEM. **B** Created in BioRender. Göttert, S. (2025) https://BioRender.com/ulwbqe3.

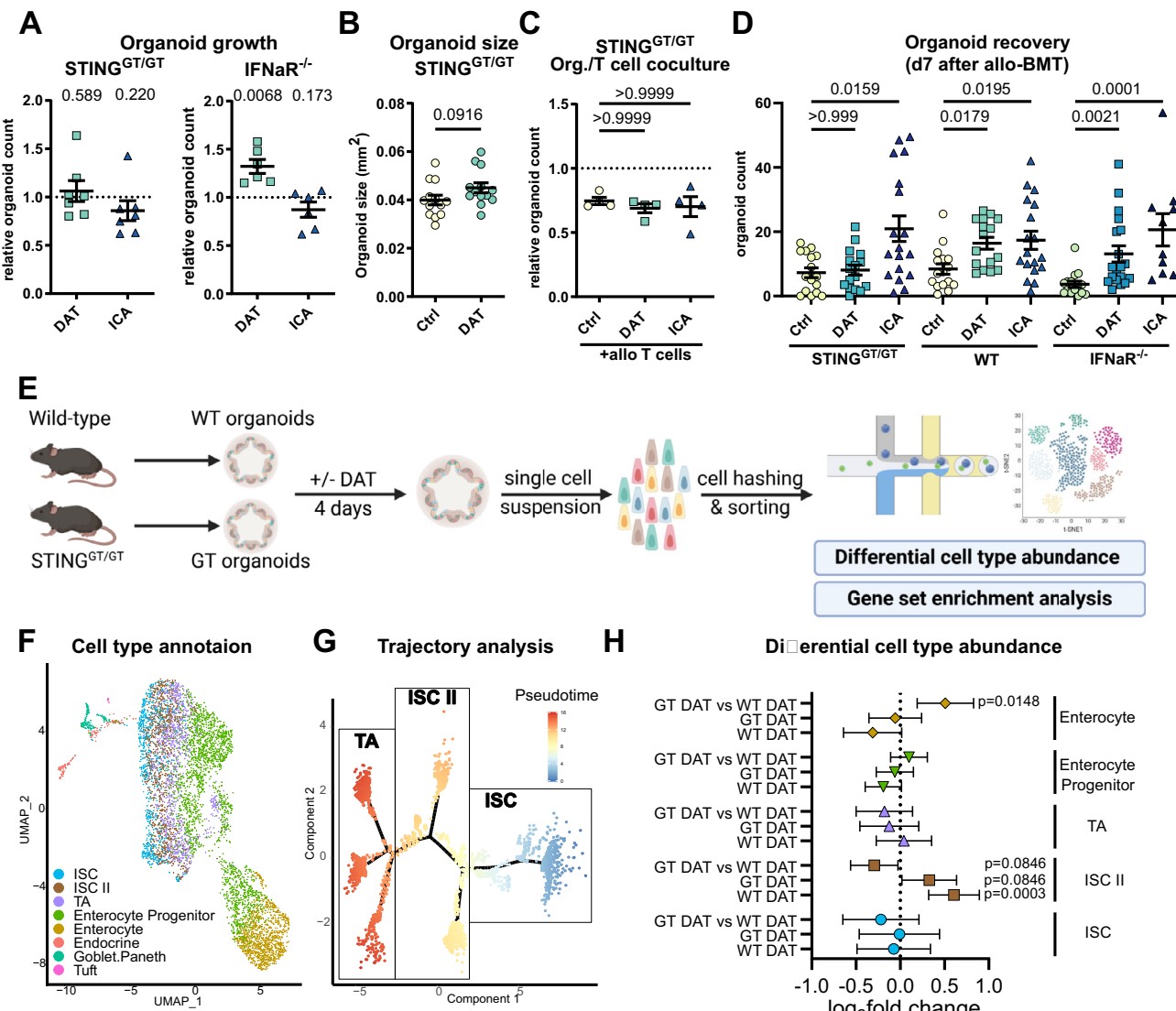

**Fig. 4 | DAT acts on the intestinal stem cell compartment and requires STING but not IFN-I signaling. A** The growth of SI organoids derived from STING^GT/GT and IFNaR^–/– mice was assessed as described in Fig. 2D. Pooled data from 6 (IFNaR^–/–) or 7 (STING^GT/GT) independent experiments. *P*-values were calculated by one sample *t*-test. **B** Small intestinal STING^GT/GT organoid size was assessed as in Fig. 2B. Pooled data from 13 independent experiments analyzed by unpaired *t*-test. **C** STING-deficient organoids were co-cultured with allogeneic activated T cells as in Fig. 2E. Pooled data from 4 independent experiments analyzed by Kruskal–Wallis test with Dunn's correction for multiple comparisons. **D** Organoid recovery was assessed in C57BL/6 WT, STING^GT/GT or IFN-I receptor deficient mice that were treated with metabolites and underwent allo-BMT as in Fig. 2I. Pooled data from 4–7 individual experiments (*n* = 15 (STING^GT/GT Ctrl and DAT, WT Ctrl and DAT), 17 (STING^GT/GT ICA and WT ICA), 18 (IFNaR^–/– DAT), 14 (IFNaR^–/– Ctrl), 10 (IFNaR^–/– ICA)). *P*-values were calculated using Kruskal–Wallis test with Dunn's correction for multiple comparisons for depicted genotypes individually. **E–H** Murine small intestinal organoids

generated from WT or STING^GT/GT mice were stimulated with metabolites as before for 4 days and subsequently analyzed by scRNA-seq. Pooled data from 3 independent experiments. **E** Experimental layout. **F** Plot of single cells in UMAP space from all experimental conditions, colored by final cell type annotation. **G** Ordering of the ISC and TA subset of cells (according to SingleR cell type annotation) on a pseudotime trajectory. **H** Differential cell type abundance analysis of DAT-treated organoids compared to their respective control or between treated WT and GT organoids, based on negative binomial regression. Data shown as log₂ fold change regression estimate with associated 95% confidence interval based on the normal distribution. *P*-values were calculated using Wald test with linear contrasts of negative binomial regression parameter estimates and adjusted for multiple testing using the FDR approach, controlling for an FDR of 10%. Data are shown as mean ± SEM if not stated otherwise. **E** Created in BioRender. Göttert, S. (2025) https://BioRender.com/v96q785.

our scRNA-seq data. In ISC, gene sets related to mTORC1 signaling and cellular stress responses were downregulated upon DAT treatment, while OXPHOS genes were downregulated in ISC-II. In contrast, GSEA of DAT-treated versus untreated STING-deficient organoids showed upregulation of genes associated with mTORC1 signaling in both ISC and ISC-II, without any downregulation of gene sets associated with metabolic activity or cellular stress responses (Fig. 5C). It must be noted that in these within-cell-type comparisons between DAT treatment and control, effects on gene expression are consistent for the

discussed gene sets with regard to the direction of regulation, but not pronounced. Thus, while DAT treatment yields a substantial expansion of the ISC-II population (Fig. 4H), which is markedly different from ISC in terms of gene set regulation (Fig. S4E), the effects of DAT treatment seen within the individual cell populations are more subtle.

We hypothesized that STING may attenuate cellular stress by counterbalancing mTORC1 signaling and metabolic activation in ISCs. STING-dependent self-renewal by autophagy can be triggered by cytosolic mitochondrial DNA (mtDNA)-sensing. OXPHOS activation

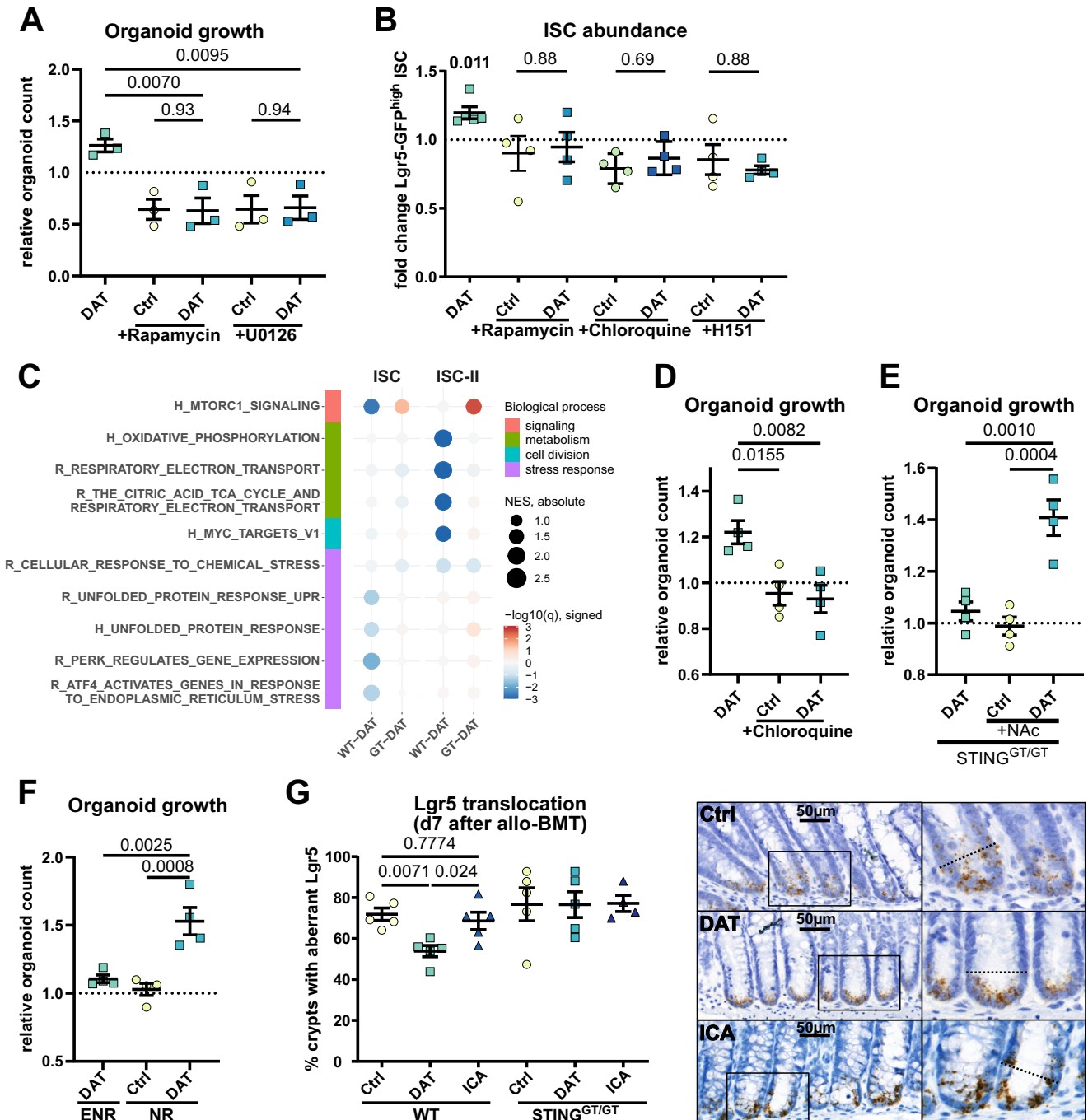

**Fig. 5 | DAT activated mTORC1 signaling in intestinal stem cells is counter-regulated by STING. A** Murine SI organoid growth was assessed as in Fig. 2D but in presence of the mTOR inhibitor Rapamycin ($1\,\mu M$) or the MAPK inhibitor U0126 ($10\,\mu M$). Pooled data from 3 independent experiments was analysed by ordinary one-way ANOVA with Dunnett's correction for multiple comparisons or unpaired *t*-test. **B** Lgr5-GFP$^{high}$ ISC abundace were assessed as in Fig. 2C, but in presence of rapamycin, the STING inhibitor H151 or chloroquine. Pooled data from 4 independent experiments. *P*-values were calculated using one sample *t*-test or Mann-Whitney U test. **C** Dotmap of GSEA results of selected pathways/gene sets for ISC and ISC-II cells and different conditions (vs. control, cell types as described in Fig. 4F, G). Dots are colored by the negative log$_{10}$ of the GSEA *q*-value (FDR), the sign indicates the direction of the regulation (up positive, down negative). The size of the dots corresponds to the normalized enrichment score (NES). Gene sets/ pathways are derived from the Hallmark (H) and Reactome (R) gene set collections of MSigDB. **D** Murine SI organoid growth was assessed as in Fig. 2D but in presence of the autophagy inhibitor chloroquine ($10\,\mu M$). Pooled data from 4 independent experiments. **E** STING deficient SI organoids were stimulated as in Fig. 4A but in presence of the antioxidant agent N-acetyl-cystein ($2.5\,mM$). Pooled data from 4 independent experiments. **F** Murine SI organoid growth was assessed as in Fig. 2D in ENR media (containing EGF, Noggin and R-spondin) or NR (without EGF). Pooled data from 4 independent experiments. **G** C57BL/6 WT and STING Goldenticket mice underwent metabolite treatment and allo-BMT as in Fig. 4D. The localization of *Lgr5* expression was assessed in large intestinal crypts by in-situ hybridization. Percentage of crypts with Lgr5 expression in the upper crypt (above +4 position) and representative images are shown. Data from 2 independent experiments (*n* = 5). *P*-values were generated by ordinary one-way ANOVA with Dunnett's correction for multiple comparisons (**A, D, E, F, G**) or unpaired *t*-test (**A**). All data are shown as mean ± SEM.

and ROS production have been associated with mitochondrial DNA (mtDNA) release into the cytosol following metabolic activation[49–55]. To test if autophagy is required for DAT's beneficial effect, we inhibited autophagy using chloroquine, which effectively abolished DAT-induced organoid growth (Fig. 5D) and decreased the abundance of Lgr5[high] ISCs in response to DAT. Likewise, direct inhibition of mTOR (with rapamycin) or STING (with H-151, a small-molecule STING inhibitor[56]), resulted in a decrease of Lgr5[high] ISCs (Fig. 5B).

As stated above, genes associated with OXPHOS are down-regulated in ISC-II cells from DAT-treated WT organoids when compared to control, according to GSEA. This is not the case in ISC-II cells from DAT-treated, STING-deficient organoids. We hypothesized that increased ROS due to higher OXPHOS might contribute to the loss of DAT's growth-promoting effect in STING-deficient organoids. Thus, we treated those with the antioxidant N-acetyl-cysteine (NAc). Reducing ROS levels restored their responsiveness to DAT treatment (Fig. 5E).

ISC proliferation is crucial for epithelial regeneration following tissue damage. Within intestinal crypts, epidermal growth factor (EGF) is a major regulator of proliferation[57–59]. EGF drives regeneration upon irradiation, and mTORC1 agonists can support this[60,61]. Given that DAT promotes ISC proliferation via mTORC1 signaling, we investigated whether DAT treatment could compensate for the absence of EGF. Therefore, we evaluated the growth-promoting effect of DAT under normal growth conditions in ENR media (containing EGF, Noggin, and R-spondin) (Fig. 2D) or EGF-depleted NR media. DAT-induced organoid growth was significantly enhanced in EGF-reduced culture conditions (Fig. 5F). Following additional depletion of endogenous EGF (which can be produced by the organoids themselves) by anti-EGF antibodies, DAT exhibited only minor compensatory capacity (Fig. S5A-B). Given DAT's capacity to foster proliferation, particularly under growth factor-compromised conditions, we further evaluated its impact on large intestinal proliferation in vivo in steady-state or after induction of conditioning damage following TBI (Fig. S5C). We observed only a minor difference in the number of Ki67+ proliferating cells in steady-state and a rapid drop in proliferating cells following irradiation. Treatment with DAT preserved proliferation both on day 1 and day 3 after irradiation. Compared to wild-type mice, this effect was compromised but not fully abolished in STING-deficient mice (Fig. S5C).

Inflammation can induce ISC exhaustion and their transition into non-functional Paneth cells, characterized by aberrant *Lgr5* localization to the "upper crypt" (above +4 position) of intestinal crypts[62]. Since allo-BMT is marked by severe intestinal inflammation, we investigated if aberrant *Lgr5* occurred in the upper crypt and whether DAT could promote the retention of Lgr5+ stem cells in the niche. We conducted *Lgr5* in situ hybridization in large intestines on day seven after allo-BMT and assessed the distribution of *Lgr5* expression in intestinal crypts. Aberrant *Lgr5* signal was detectable in most crypts of carrier-treated mice (Fig. 5G). However, in DAT-treated mice, aberrant *Lgr5* expression was significantly reduced. In STING-deficient mice, we observed no effect of DAT on the localization of *Lgr5* expression.

ISCs are a primary target of irradiation-induced intestinal damage. Accordingly, we evaluated the abundance of Lgr5+ ISCs in steady state and following TBI. We found no alteration of ISC abundance in steady-state by DAT treatment, but increased abundance on day 1 and day 3 after damage-induction (Fig. S5D). This was further accompanied by a reduced occurrence of cleaved Caspase3+ apoptotic cells (Fig. S5E). Both effects were abrogated in STING-deficient mice. In contrast, STING-deficiency had no impact on ISC abundance or induction of apoptosis compared to wild-type control.

In sum, scRNA-seq revealed that DAT promotes expansion of ISC-II cells marked by increased mTORC1 signaling. Blocking mTORC1 halted this expansion, highlighting its key role. In STING-deficient organoids, ISC-II expansion was impaired, with persistent mTORC1 activity and failure to suppress OXPHOS genes. Restoring ROS balance

rescued DAT-driven growth, indicating STING's role in regulating stress responses critical for regeneration. In vivo, DAT enhanced ISC proliferation after injury—but not at baseline—and this effect was reduced in STING-deficient mice, confirming a stem cell-intrinsic role for STING in DAT-mediated tissue repair.

## DAT promotes growth in healthy donor and patient-derived human organoids

To validate the efficacy of DAT treatment in humans (Fig. 6A), we treated LI organoids from healthy volunteers. Like in murine organoids, DAT treatment promoted organoid growth (Fig. 6B). We next wondered if the efficacy on organoid growth of DAT was compromised in patients receiving allo-HSCT. Utilizing patient-derived organoids (PDOs) derived from intestinal biopsies of either SI or LI of allo-HSCT patients, we found that DAT's growth- and size-promoting effect remained intact (Fig. 6C, D).

To confirm that the impact of DAT required intact STING signaling in humans, we inhibited STING in SI and LI PDO via H151, followed by DAT treatment. H151-treated PDOs grew poorly compared to control organoids, suggesting a requirement for STING in organoid growth. Like in murine STING[GT/GT] organoids, DAT treatment was ineffective after H151 inhibition (Fig. 6C).

We confirmed the protective effect of DAT in indirect organoid/T cell co-cultures in both SI and LI PDOs (Fig. 6E). This effect was present despite elevated IFNγ concentrations in DAT-treated co-cultures (Fig. 6F).

These data conclude that DAT exerts STING-dependent protective effects in the intestinal epithelium of healthy volunteers and remains effective in patients receiving allo-HSCT.

## DAT treatment only moderately impacts T cell activation

Besides intestinal barrier protection, mechanisms that may contribute to DAT's effect include (i) increased frequency of donor regulatory T cells (Tregs), (ii) impaired function of tissue-damaging alloreactive CD8+ effector cells, or (iii) reduced capacity of host antigen-presenting cells (APCs) to promote T cell alloreactivity. Given that we found elevated IFNγ concentrations in DAT-treated human organoid/T cell co-cultures (Fig. 6F), we stimulated CD3+ T cells from peripheral blood mononuclear cells (PBMC) of healthy volunteers with anti-CD3/CD28 beads for 24 h, and added metabolites in escalating doses (Fig. 7A).

Direct treatment with the metabolites DAT or ICA resulted in opposing effects: while the ratio of CD4+ to CD8+ T cells remained unaffected by either metabolite (Fig. S6B), DAT, but not ICA treatment, reduced the frequency of Tregs in a dose-dependent manner (Fig. 7B). The frequency of CD4+ T cells expressing the inflammatory cytokine interferon γ (IFNγ) was not affected by either metabolite (Fig. S6C). Only after applying unphysiologically-high DAT doses (1000 µM, henceforth referred to as "high-dose DAT") did we observe increased IFNγ expression per cell by mean fluorescence intensity (MFI). ICA had the opposite effect and suppressed IFNγ expression (Fig. 7C). Consistent with increased activation, high-dose DAT promoted the immune checkpoint receptor PD-1 expression on CD4+ T cells (Fig. S6D).

On CD8+ T cells, neither low nor high-dose DAT treatment increased the abundance of IFNγ producing cells or the expression levels of IFNγ or PD-1 (Figs. 7D, and S6E, F). Instead, we observed reduced IFNγ expression following low-dose DAT treatment (Fig. 7D). ICA treatment lowered the abundance of IFNγ+CD8+ cells and the expression of IFNγ and PD-1 (Figs. 7D, and S6E, F) in a dose-dependent manner. These results indicate that only high-dose DAT can promote an effector phenotype in human T cells. This was corroborated by comparing the ratio of regulatory to effector T cells: high-dose DAT resulted in significantly reduced ratios of Tregs to effector CD4+ (Fig. 7E) and CD8+ (Fig. 7F) T cells. We observed a similar reduction in Treg to effector T cell ratios in DAT-treated murine splenic CD4+ and CD8+ T cells (Fig. S6G, H).

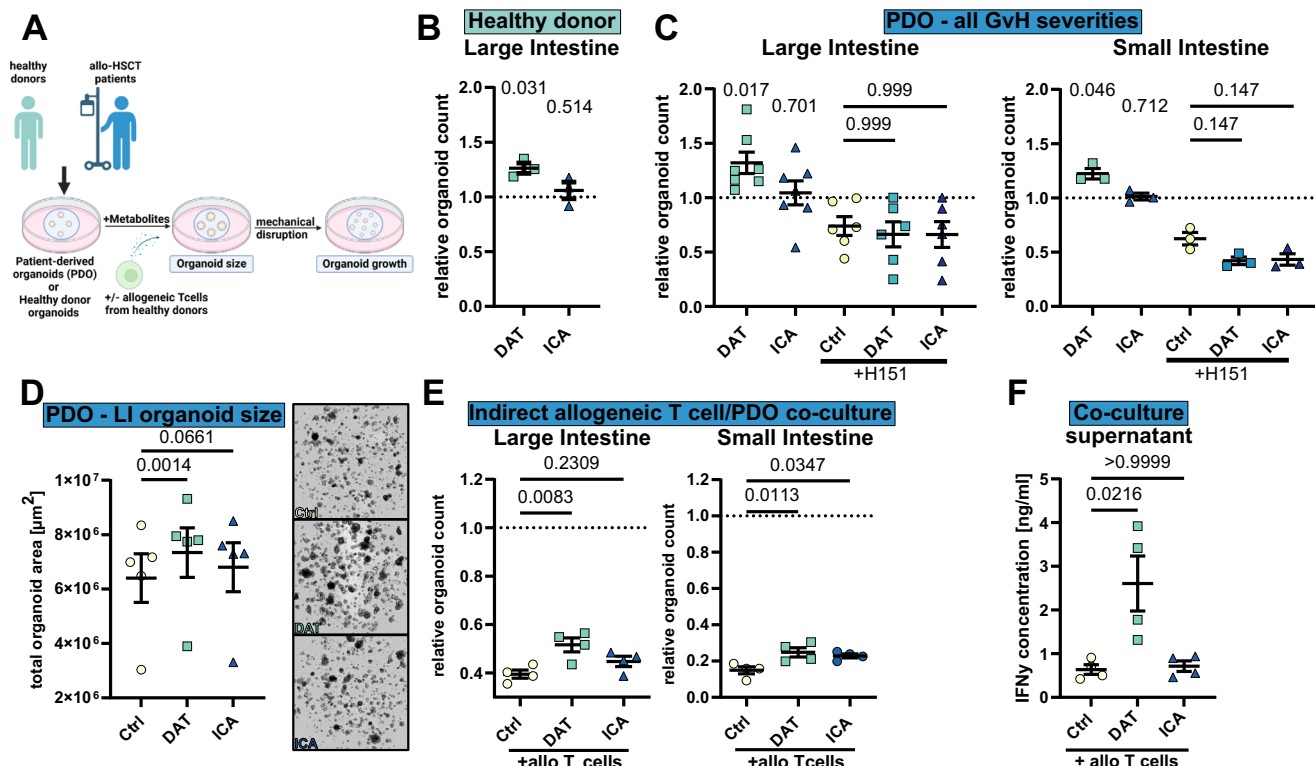

**Fig. 6 | DAT treatment is effective in healthy donor and patient-derived human organoids. A** Experimental overview. **B** Human LI organoids derived from healthy donors were stimulated with metabolites and their growth assessed. Pooled data from 3 different donors in independent experiments. **C** Human LI and SI organoids derived from patients receiving allo HSCT were stimulated with metabolites as in (**A**) or in presence of H151. Pooled data from 3 (SI), 7 (LI Ctrl, DAT, ICA) or 6 (LI H151, H151 DAT, H151 ICA) donors in independent experiments. **D** Size of LI PDO after 6 days of metabolite treatment assessed as total organoid area. Pooled data from 5 independent experiments. **E** Human organoid/T cell co-culture without direct interaction. To mimic cytokine-mediated toxicity, LI PDO were co-cultured with CD3/28 beads + 30 U/ml IL-2 activated pan-T cells derived from healthy volunteers in the surrounding media for 4 days and their growth assessed as in (**A**). Counts are shown relative to untreated organoids. Pooled data from 4 different patients in independent experiments. **F** IFNγ concentration in cell culture supernatants of organoid/T cell co-cultures assessed by cytokine bead array. $N = 4$ wells from two donors. *P*-values were calculated by one-sample *t*-test (**B**+**C**), ordinary one-way ANOVA with Dunnett's correction for multiple comparisons (**E**+**F**) or paired one-way ANOVA with Dunnett's correction for multiple comparisons (**D**). All data are shown as mean ± SEM. **A** Created in BioRender. Göttert, S. (2025) https://BioRender.com/6quwbnv.

As the GvL effect is mediated in part by T cell cytotoxicity, we next assessed effector CD8+ T cell function by quantifying Granzyme B expression. Only high-dose DAT treatment increased the abundance of Granzyme B+ CD8+ T cells and Granzyme B expression. Conversely, ICA had the opposite effect (Fig. 7G, H), suppressing cytotoxic protein expression.

These findings suggest that high-dose DAT treatment may risk exacerbating GvHD. To assess this, we treated mice with high DAT doses by supplying it *ad libitum* via the drinking water (corresponding to ~3400 mg DAT/kg body weight per day) from day −7 either until the day of allo-BMT (day 0) or continuously throughout the experiment (Fig. 7I). Indeed, continuous, high-dose DAT treatment induced hyperacute GvHD (Fig. 7J). Compared to gavage of low-dose DAT (170 mg DAT/kg body weight per day), high-dose DAT administered until day 0 was also less effective at preventing GvHD.

We then utilized the murine allo-BMT model (Fig. 1D) to assess the impact of low-dose DAT or ICA on murine donor T cells in vivo. To this end, we analyzed donor (H-2K$^{b+}$) T cells isolated from the intestinal epithelium at day +7 after allo-BMT (Fig. S7A). Metabolite treatment did not affect epithelial donor CD4+ T cells (Fig. S7B) and we did not observe any effects on Treg abundance following DAT or ICA. However, DAT treatment resulted in a moderately increased abundance of IFNγ+CD8+ T cells and expression of PD-1 (Figs. 7K, and S7B) and aligns with a previous report showing that DAT can directly activate murine TCR-stimulated CD8+ T cells in vitro[32]. In contrast, ICA treatment trended towards a reduced frequency of IFNγ+CD8+ T cells and elevated PD-1 expression.

Finally, we could not detect any significant effects following stimulation with low-dose DAT on human (Fig. S6I–L) or murine APCs[32].

Our data indicate that DAT at physiological doses (i.e., low-dose, corresponding to fecal concentrations in patients receiving allo-HSCT) only moderately impacted T cells. Therefore, its beneficial effect on GvHD is unlikely to be mediated by Treg promotion, suppression of allogeneic T cells, or APC priming of alloreactivity. In the context of GvHD, DAT's T-cell-dependent effect appeared to be secondary to its robust effect on epithelial cells, while the dose-dependent amplification of both human and murine CD8+ T cells may partially explain DAT's impact on GvL (Fig. 1I).

## Discussion

In this study, we demonstrate that the microbial metabolite DAT, identified in patients receiving allo-HSCT and associated with improved overall survival and reduced relapse at two-year follow-up, concurrently mitigates GvHD and contributes to anti-tumor immune responses in murine in vivo and human ex vivo models. This "dual function" of DAT is unique among other microbial metabolites in allo-HSCT and arises from DAT's activating effects on T cells and its protective role in maintaining the intestinal barrier (summarized in Fig. 8, upper panel):

There is robust evidence that the intestinal microbiome impacts the outcome following allo-HSCT and the incidence of

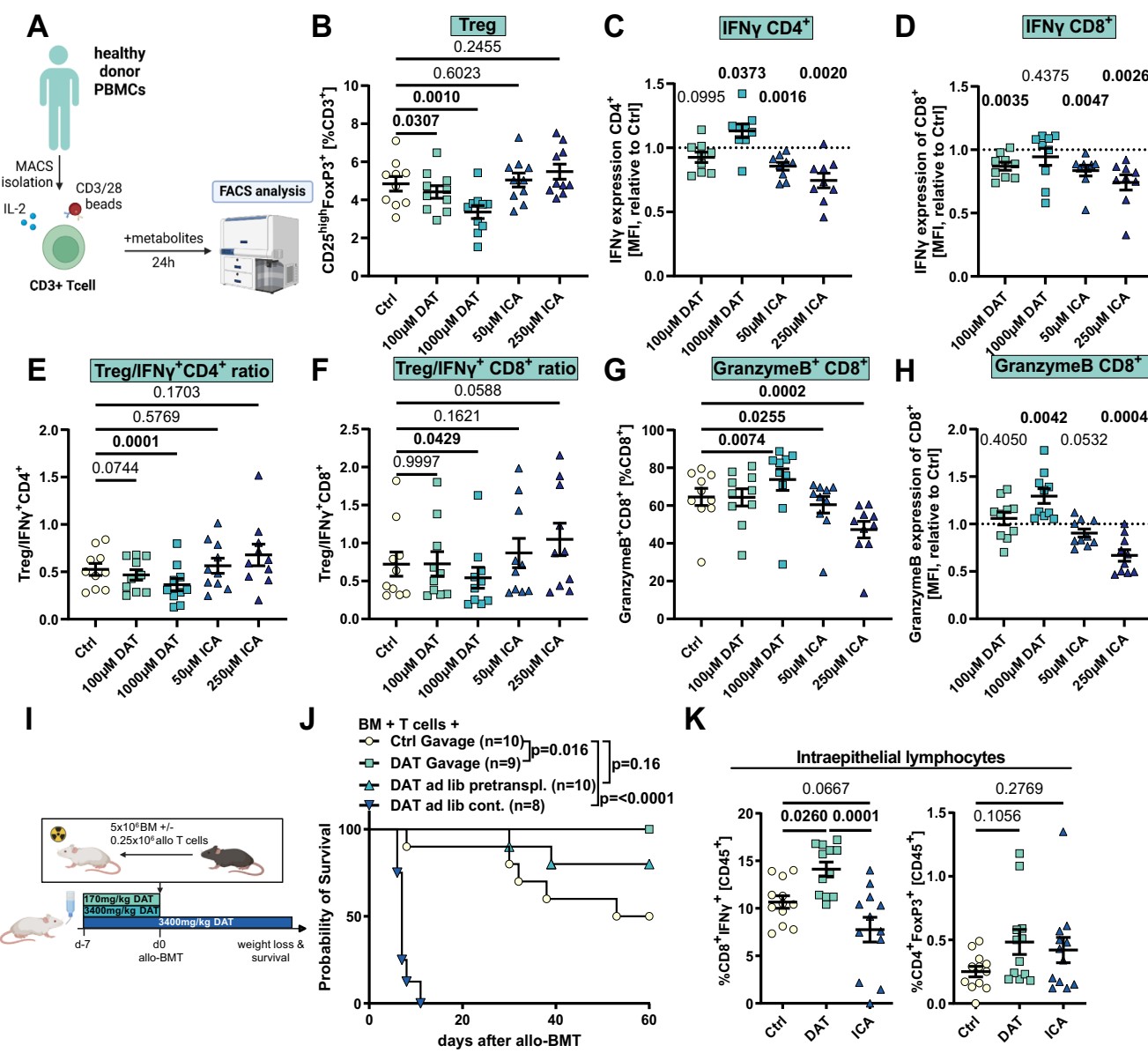

**Fig. 7 | DAT promotes T cell activation in vitro in humans and in vivo in mice.**
**A** Experimental overview. Human pan-T cells were stimulated with human anti-CD3/28 beads, 30 U/ml hIL-2 and indicated concentrations of metabolite for 24 h and analyzed by flow cytometry (Fig. S6A). Abundance of the indicated T cell subpopulations (**B–G**), protein expression according to mean fluorescence intensity (MFI) normalized to the respective control (**C**, **D+H**) or ratio of Tregs to IFNγ+CD4+ or IFNγ+CD8+ T cells (**E**, **F**). Pooled data from 10 healthy individuals. Data was analyzed using ordinary one-way ANOVA for paired data with Dunnett's correction for multiple comparisons. **I** Experimental in vivo design. **J** Survival of mice. Ctrl and DAT gavage mice are also shown in Fig. 1E. Data was analyzed by Log-Rank test from 2 independent experiments. **K** BALB/c mice were orally treated with the metabolites and underwent allo-BMT as depicted in Fig. 1D. On day 7 mice were sacrificed and H-2Kᵇ⁺ donor T cells were analyzed by flow cytometry (Fig. S7A). Abundance of IFNγ+CD8+ cytotoxic T cells and abundance of regulatory T cells. Pooled data from 3 independent experiments and *n* = 12 mice. Data was analyzed using ordinary one-way ANOVA with Dunnett's correction for multiple comparisons. If not depicted otherwise, all data are shown as mean ± SEM. **A** Created in BioRender. Göttert, S. (2025) https://BioRender.com/wzp97wb. **I** Created in BioRender. Göttert, S. (2025) https://BioRender.com/utm6phc.

GvHD[13]. The perturbation of the microbiome due to antibiotic use and the adverse consequences of the ensuing dysbiosis are well documented[5,6,9,14,15]. Consequently, efforts are underway to restore microbial diversity after allo-HSCT, i.e., via FMT, with promising outcomes[63–65]. Similarly, microbiota-derived metabolites are associated with positive outcomes following allo-HSCT[19–22]. Specific metabolites, such as SCFAs (e.g., butyric acid), bile acids, and tryptophan-derived metabolites like ICA, shape the intestinal immune cell compartment towards an anti-inflammatory environment by promoting the expansion of Tregs, releasing anti-inflammatory and regenerative cytokines such as IL-10 and IL-22, and suppressing effector T cells and APCs[28,30,66–72].

Consistently, we observed that ICA prevented T-cell toxicity in the co-culture model of in vitro GvHD, aligning with recent reports indicating partial protection from GvHD-related mortality by late (i.e., after day +12) ICA application[28]. We attribute this effect to an immunosuppressive role of ICA in allo-BMT, evidenced by reduced T cell activation and downregulation of MHC II expression in epithelial cells and monocytes.

In contrast, DAT enhanced effector T cell phenotypes in vivo and in vitro, especially at high doses, which aligns with previous studies showing that high-dose DAT improves ICB efficacy in murine tumor models[32] and fosters protection against influenza, Covid-19 and *Salmonella enterica* infections[33,34,73]. However, high-dose DAT posed a risk

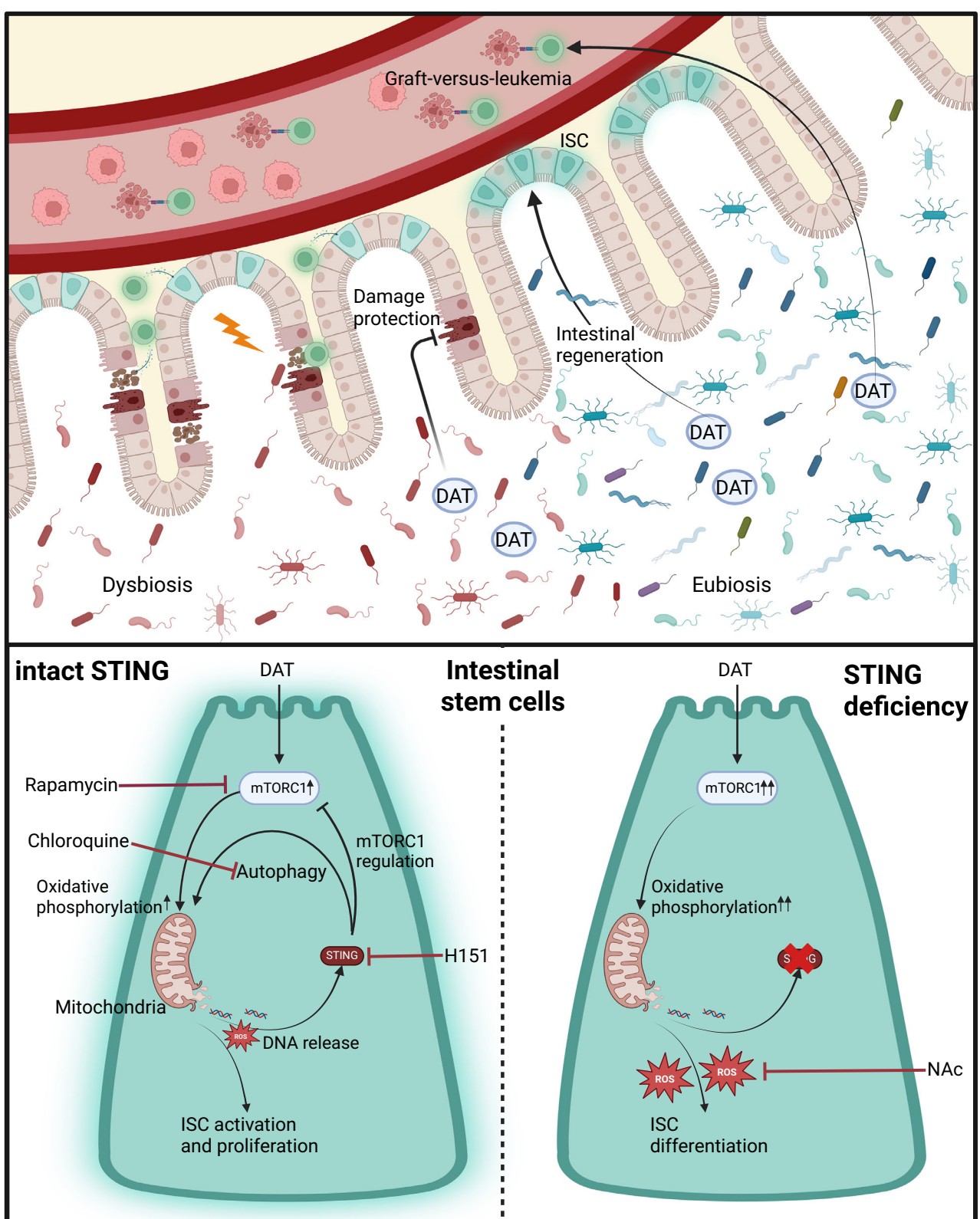

**Fig. 8 | Graphical summary.** (Upper panel) The microbial metabolite DAT, produced by a healthy microbiome or supplemented as a synthetic metabolite, protects the intestinal epithelium from damage and promotes intestinal regeneration via intestinal stem cells. Systemically, it promotes effector function of T cells and promotes graft-versus-leukemia responses. (Lower panel) In ISCs, DAT promotes proliferation and metabolic activation, but requires STING for metabolic regulation. In the absence of STING, the ability to react to metabolic stress is compromised and ISCs undergo ROS-mediated differentiation. Created in BioRender. Göttert, S. (2025) https://BioRender.com/p48p038.

for hyperacute GvHD in the allo-BMT mouse model, likely due to excessive stimulation of allogeneic T cells. Conversely, low-dose DAT protected from GvHD and produced only a moderate increase in murine CD8+ T cells in vivo. This may partly explain the capacity of low-dose DAT to significantly delay Ba/F3 FLT3-ITD-Luc acute leukemia progression without compromising GvHD protection. DAT had only minor effects on human APCs, aligning with previous reports that DAT promotes APC activation only in combination with viral co-stimulation but not on its own[32,33].

Compared to the moderate effects of DAT on immune cells, low-dose DAT was very potent in stimulating intestinal epithelial cells. DAT increased the epithelial turnover in human and murine organoids, expanded the ISC compartment, and improved resistance to irradiation and cytokine-mediated toxicity. In vivo, DAT protected the intestinal barrier by suppressing the induction of apoptosis, preventing the loss of ISCs, enhancing intestinal proliferation after TBI, and promoting intestinal regeneration. Suppression of p53-driven apoptosis and ISC loss is a well-established protective mechanism in intestinal injury, as demonstrated with various interventions, including growth factors and pharmacological agents such as GSK-3 inhibitors[74–76]. Similarly, protecting the intestinal barrier from conditioning chemo- or radiotherapy and T cell toxicity is an effective mechanism to prevent early allogeneic T cell priming and subsequent GvHD[25,28,36]. Additionally, DAT prevented the thinning of the intestinal mucus layer after allo-BMT, which has been associated with aggravated GvHD[43]. Our data, therefore, indicate that at low physiological doses, DAT's robust protective effect on the intestinal epithelium is balanced with moderate T cell activation, resulting in a benefit for both GvHD and GvL.

We also aimed to assess whether DAT treatment remained effective without an intact microbiome or during antibiotics-induced dysbiosis. DAT's protective effect on ISCs was compromised in GF mice in vivo. In contrast, ex vivo, DAT treatment was effective in both GF and FMT-colonized GF mice-derived crypts, consistent with data showing that organoids maintain a robust transcriptome/proteome phenotype independent of GF status[77]. We conclude that the in vivo effect of DAT treatment is relying on additional microbial signals (e.g., microbe-associated molecular patterns) and physiological intestinal development, which are compromised in GF mice[40]. Along these lines, the ability of DAT to promote organoid regeneration was restored following gut decontamination with ampicillin/enrofloxacin in SPF mice.

Intestinal dysbiosis is common in patients receiving allo-HSCT and is associated with GvHD and broad-spectrum antibiotics, such as meropenem. Despite meropenem treatment, both DAT and ICA still induced tissue regeneration in the mouse allo-BMT model. Previously, we showed that broad-spectrum antibiotics worsen the efficacy of ICB in murine solid tumor models, but DAT treatment could restore efficacy[32]. Together, these data suggest that endogenous DAT, produced by anaerobic microbiota such as *F. plautii*–identified in murine and allo-HSCT patient studies[19,33], may contribute to steady-state tissue regeneration and anti-tumor immunity.

FMT is the most advanced among the different microbiome-based therapies currently under development, including for the treatment of acute, steroid-refractory GvHD[78]. Intriguingly, FMT of the flavonoid-degrading, DAT-producing human-isolated *F. plautii* significantly improved organoid regeneration following allo-BMT and broad-spectrum antibiotics-induced dysbiosis. Thus, a "precision" FMT using defined metabolite-mixtures or metabolite-producing bacterial strains may be a potent alternative to synthetic metabolites metabolites, able to supply DAT endogenously at the interface between microbiota and host epithelium.

Our study finds that DAT-induced intestinal organoid growth is dependent on mTORC1 signaling. mTORC1 plays an essential role in intestinal homeostasis and regeneration and regulates metabolic activation and proliferation in response to growth factors like EGF[48,79–81]. Its transient activation in ISCs after damage is crucial for regeneration[61,82] and beneficial effects associated with fasting[47,48,83]. Consistently, we observed that DAT treatment in WT organoids expanded ISCs for which transcriptomic analysis indicated elevated metabolic and proliferative capacity, while retaining expression of the stem cell markers Lgr5 and Olfm4 (ISC-II).

The STING pathway, known for its role in innate immunity[84,85], recognizes microbial and non-microbial ligands like cyclic di-nucleotides and free double-stranded (ds) DNA. These ligands, derived from intestinal microbiota or other sources, are constantly sensed by STING and induce tonic signaling and IFN-I responses, influencing intestinal homeostasis and local and systemic immunity[86–92]. Synthetic direct cGAS/STING agonists like dsDNA can stimulate IFN-I responses to protect the intestinal epithelium and against GvHD in mice[36], even though the cellular targets, its associated mechanisms and the relevance in humans remain unclear. Since the DAT phenotype can be restored in STING-deficient organoids by reducing ROS, our data suggest that DAT does not directly activate STING in ISCs. Instead, it appears to utilize STING for metabolic regulation, thereby maintaining an undifferentiated, regenerative stem cell state. Consequently, other than in WT organoids, DAT induced the upregulation of genes associated with mTORC1 signaling in STING-deficient organoids, indicating a possible dysregulation of mTORC1 signaling in the absence of STING. Dysregulated mTORC1 activation has been associated with depletion of the ISC pool as well as stem cell exhaustion following repetitive stress and aging, stem cell differentiation, and intestinal barrier damage induced by a Western diet[93–96]. Furthermore, STING's regulatory function occurs independently of IFN-I signaling, consistent with previous reports in other cell types showing that STING can recognize the release of mtDNA following stress, thus contributing to metabolic regulation, and reports of elevated STING expression in ISCs[54,55,97]. An ISC-intrinsic role of STING is further supported by our in vivo observation that the protective effect of DAT on ISCs following TBI is attenuated in STING-deficient mice despite preserved induction of proliferation. However, while STING deficiency compromised the beneficial effects of DAT treatment, it did not impact the direct response to irradiation damage compared to wild-type mice. This finding extends earlier reports implicating STING-mediated intestinal regeneration by IFN-I induction following TBI[36,98]. We therefore propose the following model (summarized in Fig. 8, lower panel): In WT organoids, DAT promotes ISC proliferation and metabolic activation via mTORC1 signaling and OXPHOS. ROS, a byproduct of OXPHOS, triggers mitochondrial stress and STING activation. Upon activation, STING provides a negative feedback loop that attenuates metabolic stress by regulating mTORC1 and promoting autophagy[49–53], increasing the ISC-II pool and limiting their further differentiation. As a result, DAT promotes a regenerative phenotype in organoids (Fig. 8, lower panel (left)). In contrast, STING deficiency results in a compromised ability to react to metabolic stress, unchecked mTORC1 signaling and ROS production. Consequently, ISCs undergo differentiation[99], and the capacity for organoid regeneration is abolished (Fig. 8, lower panel (right)).

Taken together, we identify a microbial metabolite (DAT) with beneficial, dual functions on GvHD and GVL in humans and mice and uncover a link between microbial metabolites, mTORC1, and the STING pathway to induce regulation of the ISC compartment.

From a translational perspective, restoring intestinal DAT levels, e.g., by exogenous administration of "pure" metabolites, flavonoid-degrading probiotics (e.g., *F. plautii*), or "precision" FMT products with defined metabolite composition and prespecified dose ranges, represents an attractive treatment approach during the use of broad-spectrum antibiotics in patients receiving allo-HSCT. Combining DAT with other metabolites may also provide synergisms for tailoring these therapies according to specific indications, i.e., regeneration-promoting metabolites for GvHD, immunity-enhancing metabolites

for GVL and perhaps other cancer immunotherapies (e.g., chimeric antigen receptor (CAR) T cells, immune checkpoint inhibitors, bispecific antibodies). Finally, DAT's association with beneficial GvHD and GvL responses may be useful as a clinical biomarker for predicting clinical outcomes following allo-HSCT.

A limitation of the study is that although high DAT concentrations were independently associated with reduced relapse rates in patients receiving allo-SCT, we caution against using DAT as a standalone biomarker outside of the IMM-RI, as the cohort size with $n = 50$ patients precluded multivariable analysis. The association of microbial metabolites with additional clinical outcomes, including GI-GVHD, was previously reported[19].

## Methods
### Patient cohort
The research in humans reported here complies with all relevant ethical regulations. Study protocols were approved by the institutional review boards (IRB) "Ethikkommission der TU München" (295/18 S) and "Ethikkommission der Universität Regensburg" (14-47_1-101, 14-101-0047, 21-2521-01). The study is registered at the German Clinical Trials Register (DRKS00034175).

All patients receiving allo-SCT at the University Hospital of the Technical University Munich (between 2019 and 2021) and University Hospital Regensburg (between 2018 and 2021) were screened. Patients were enrolled after obtaining informed consent and in accordance with the IRB-approved study protocols referenced above. Patients did not receive financial participant compensation.

We report the results of an observational, prospective, longitudinal clinical trial. Patients receiving allo-HSCT were treated per standard-of-care.

The study cohort included female and male patients. Sex (biological attribute) was assigned based on electronic patient records. We did not consider sex in the study design and did not perform separate analysis according to sex, since sex did not affect microbiome-predicted outcomes in other cohorts of allogeneic stem-cell transplantation patients[13,100]. At the University Hospital Regensburg, ethnicity was determined by self-reporting, all patients identified as "Non-Hispanic/white".

Fecal samples were collected once a week, beginning on day −7 before allo-SCT and ending on day +21 after allo-SCT. Samples were obtained on one to two days around the specified time points. Stool samples were collected as unprocessed, native samples and transferred to the laboratory within 4 h, where they were stored at −80 °C until further processing.

Patients with a minimum of three longitudinal samples were included in the analysis. No statistical method was used to pre-determine sample sizes, but the size of our cohort is similar to those reported in previous publications. Randomization did not apply in our non-interventional cohort. Data collection was not performed blind, and investigators were not blinded to outcome assessment.

### Quantification of fecal metabolites in human samples
Metabolite levels at days −7 and +7 to +21 relative to allo-HSCT were used for the analysis. When more than one sample per patient was available, the mean metabolite concentration was used. Patients were stratified into high and low DAT and ICA groups, respectively. The threshold concentrations for these group assignments were determined as previously reported[19].

### Analysis of clinical outcomes
Overall survival at a follow-up of two-years was analyzed using the Kaplan–Meier estimator. Cumulative incidence of relapse was estimated accounting for its competing risk TRM. Cumulative incidence functions between groups were tested for equality using Gray's test[101] and the hazard ratio for the different endpoints was calculated in R based on the Fine and Gray model using the crr function[102] and the Cox regression model (OS) using SPSS v29.

### Mice
C57BL/6J and BALB/c mice were purchased from Janvier-Labs. Mice genetically modified for the interferon-α receptor 1 (B6(Cg)-Ifnar1tm1.2Ees/J; RRID: IMSR_JAX:028288; here referred to as IfnaR$^{−/−}$), STING (C57BL/6J-Sting1gt/J; RRID:IMSR_JAX:017537; here referred to as STING Goldenticket, STING$^{GT/GT}$), MAVS (B6;129-Mavstm1Zjc/J; RRID:IMSR_JAX:008634; here referred to as MAVS$^{−/−}$), Lgr5-GFP (B6.129P2-Lgr5tm1(cre/ERT2)Cle/J; RRID:IMSR_JAX:008875) STING flox (B6;SJL-Sting1tm1.1Camb/J, RRID:IMSR_JAX:031670) and Vilin-Cre (B6.Cg-Tg(Vil1-cre)997Gum/J, RRID:IMSR_JAX:004586) have been described previously[103–106] and were breed in the animal facilities of the University Hospital Regensburg or Technical University Munich. Germfree BALB/c mice were bred and maintained in special isolators at the GF mouse facility of the University of Regensburg and transferred to individual sterile isocages at the beginning of experiments. Transplantation and treatment with metabolites were performed under sterile conditions in a laminar flow. For irradiation, mice were transferred under sterile conditions to a sterile HEPA filter box, which was additionally tape-sealed to prevent contamination. GF status was confirmed at the beginning and the end of each experiment. The handling of germ-free mice was performed by specifically trained staff.

Animals were housed in SPF animal research facilities and monitored for pathogens according to FELASA recommendations. Mice were between 6 and 10 weeks old at the beginning of the experiments. Mice were maintained on a 12-h light–dark cycle, a constant temperature of 24 °C and a humidity level of 55%–65%. Ad libitum access to autoclaved standard laboratory chow (V1124-300, Ssniff) and water was guaranteed at any time. Environmental enrichment, including nesting material, houses, and chew toys, was provided to promote their welfare and well-being. The animals were housed in the facility where the experiments took place for at least one week prior to the start of an experiment for acclimatisation. Genetically modified mice were co-housed with age- and sex-matched WT mice for a period of at least 4 weeks before starting in vivo experiments. Treatment groups were not co-housed to avoid microbial/metabolite transfer through coprophagy between groups. Animal studies were approved by the local regulatory authorities (Regierung von Oberbayern, ROB-55.2-2532.Vet_02-15-09, ROB-55.2-2532.Vet_02-20-168 or Regierung von Unterfranken, RUF-55.2.2-2532-2-1301).

### Media and reagents
RPMI-1640 (Invitrogen) was supplemented with 10% (v/v) FCS (Gibco), 2 mM L-glutamine, 100 μ/ml penicillin and 100 μg/ml streptomycin (Sigma–Aldrich). Complete RPMI contained 10% (v/v) FCS, 2 mM l-glutamine, 100 μ/ml penicillin, 100 μg/ml streptomycin, 50 μM β-mercaptoethanol (Gibco), 1× non-essential amino acids (Gibco), 1 mM sodium pyruvate (Gibco) and 10 mM HEPES (Gibco). Human T cell media contained 10% (v/v) FCS, 2 mM l-glutamine, 100 U/ml penicillin, 100 μg/ml streptomycin, 1 mM sodium pyruvate (Gibco), 10 mM HEPES (Gibco) and was supplemented with 30U/ml hIL-2 (Peprotech).

ENR media contained advanced DMEM/F12 (Life technologies), 2 mM L-glutamine (Sigma), 10 mM HEPES (Life technologies), 100 μ/ml penicillin, 100 μg/ml streptomycin (Life technologies), 1.25 mM N-acetyl cysteine (Sigma), 1× B27 supplement (Life technologies), 1× N2 supplement (Life technologies), 50 ng/ml mEGF (Peprotech), 100 ng/ml rec. mNoggin (Peprotech), 10% human R-spondin-1-conditioned medium of hR spondin-1 transfected HEK 293T cells. In the indicated experiments, EGF-depleted ENR (indicated as NR media) was used that contained all mentioned ingredients besides EGF. Additional blocking of organoid-produced EGF was achieved by supplying 0.5 μg/ml aEGF antibody (Catalog # AF2028, Bio-Techne). 2.5 mM N-acetyl-cystein was used in NAc high ENR media. WENR

contained advanced DMEM/F12 (Life Technologies), 2 mM L-glutamine (Sigma), 10 mM HEPES (Life Technologies), 100 µl/ml penicillin, 100 µg/ml streptomycin (Life Technologies), 1.25 mM N-acetyl cysteine (Sigma), 1× B27 supplement (Life Technologies), 50 ng/ml hEGF (Peprotech), 100 µg/ml Normocin (InvivoGen), 50% L-WRN-conditioned medium of L-WRN cells. Human or murine IntestiCult intestinal organoid growth media (STEMCELL Technologies) was supplemented with 100 µl/ml penicillin and 100 µg/ml streptomycin.

Gavage solution of DAT was prepared with 17 mg/ml 3-(4-hydroxyphenyl)-propionic acid (DAT, 98%, Sigma–Aldrich) in 5% EtOH (Carl Roth) in water. Drinking water was prepared with 100 mM DAT and neutralized with NaOH (Merck) in tap water. The respective solvents served as controls. ICA was prepared with 15 mg/ml as described before[28]. Gavage solutions of both metabolites were thereby prepared in equimolar concentrations. Daily in vivo uptake of metabolites was estimated at 170 mg/kg body weight (DAT gavage), 3400 mg/kg body weight (DAT drinking water with 4 ml water uptake/20 g mouse) or 150 mg/kg body weight (ICA). Gavage solutions and drinking water were filtered through 0.22 µm sterile filters and were prepared freshly every 3 days. Broad-spectrum antibiotics with 0.5 g/L Ampicillin and 0.25 g/L enrofloxacin or 0.625 g/L Meropenem were administered via the drinking water (ad libitum). Drinking water supplemented with antibiotics was replaced every 2 days to maintain constant antibiotic activity.

## GvHD induction after allo-BMT

For GvHD induction after allo-BMT a major mismatch (H-2k$^d$/H-2k$^b$) with myeloablative TBI was used. BALB/c recipients were irradiated with 8 Gy and after 3–4 h intravenously injected with 5 × 10$^6$ allogeneic (C57BL/6J) T-cell-depleted bone marrow cells alone or in combination with 0.25 × 10$^6$ allogeneic T cells. Bone marrow cells were isolated from naïve C57BL/6J mice, and TCD was performed using CD90.2 MicroBeads (Miltenyi Biotec). Donor T cells were isolated from pooled spleens of naïve C57BL/6J mice using CD4 and CD8 MicroBeads (Miltenyi Biotec), respectively, and mixed in equal proportions. Recipient mice were monitored daily in a non-blinded manner and euthanized when they reached one of the human end point criteria: (1) more than 30% weight loss; (2) bloody diarrhea; (3) apathy or if other determination criteria were met according to standard legal procedure.

## Graft-versus-leukemia model

The GvL model was performed as previously described[107]. In brief, Ba/F3 cells (murine, DSMZ no.: ACC 300) transfected with the FLT3-ITD oncogene as well as the luciferase (luc)$^+$ and green fluorescent protein (GFP)$^+$ serving as marker genes were used[108]. Cells were kindly provided by Natalie Köhler. BALB/c mice received 8 Gy TBI and after 3–4 h were intravenously injected with 5 × 10$^6$ allogeneic (C57BL/6J) BM cells and 500–750 tumor cells. On day 2 post transplantation, 0.25×10$^6$ allogeneic T cells were transplanted. BM-only mice received no additional T cells. Expansion of leukemia cells was monitored by BLI.

For the FLT3-ITD/MLL-PTD AML model, C57Bl6/J mice were irradiated with 11 Gy and after 3–4 h intravenously injected with 5 × 10$^6$ allogeneic (BALB/c) BM cells and 5000–7500 tumor cells. On day 2 post transplantation, 0.25 × 10$^6$ allogeneic T cells were transplanted. BM-only mice received no additional T cells. When mice reached the human endpoint (as defined in "GvHD induction after allo-BMT") or the end of the experiment, spleen and bone marrow were analyzed by FACS to determine the engraftment of CD117$^+$H2k$^{b+}$ tumor cells to define relapse or GvHD as the cause of death.

## Bioluminescence imaging (BLI)

For in vivo BLI, mice were injected intraperitoneally (i.p.) with luciferin (150 mg/kg body weight) and narcotized by inhalation with 2.5% isoflurane. Imaging was performed with the IVIS 50 BLI system (Xenogen) and quantified using the Living image Software (Xenogen).

## Induction of intestinal damage by total body irradiation, doxorubicin or allo-BMT

Following in vivo models for intestinal damage were used:

To induce damage by total body irradiation, C57BL/6 mice were irradiated with 11 Gy and subsequently intravenously injected with 5 × 10$^6$ syngeneic (C57BL/6) BM cells.

Intestinal damage by Doxorubicin was induced in C57BL/6 mice by intraperitoneal injection of 7.5 mg/kg (d3 granulocyte infiltration assay) or 20 mg/kg (d6 organoid recovery assay) doxorubicin as described before[39].

Generally, allogeneic tissue damage was induced in C57BL/6 mice following 11 Gy TBI by intravenous injection with 5 × 10$^6$ allogeneic (BALB/c) BM cells alone or in combination with 2 × 10$^6$ allogeneic T cells. Due to increased susceptibility to intestinal tissue damage, the T cell dose was reduced to 1.5 × 10$^6$ allogeneic T cells in IFNaR$^{-/-}$ and meropenem-treated mice. Germfree BALB/c mice were transplanted with 0.4 × 10$^6$ allogeneic (C57BL/6) T cells and 5 × 10$^6$ BM cells following 8 Gy TBI.

## Fecal microbiota transfer

C57Bl/6 mice received the antibiotics ampicillin and enrofloxacin for 5 days via drinking water. On days 2 and 5 after completion of antibiotics, mice received suspensions of the indicated bacterial strain (DSM 6740, DSM 26117, German Collection of Microorganisms and Cell Cultures GmbH) or 20 mg of cecal content from SPF mice donors as control via oral gavage and rectal application. Mice received 7.5 × 10$^8$ bacteria in a 150 µl volume by 2/3 oral gavage and 1/3 rectal application. Bacterial strains were expanded under strictly anaerobic conditions in Anaerobe Basal Broth (ABB; Thermo Scientific/Oxoid # CM0957). The concentration of each bacterial species was quantified based on OD600. Allo-BMT was performed on day 7 post antibiotic treatment. Colonization was confirmed by performing qPCR using *F. plautii*-specific primers from stool pellets.

## qPCR for fecal bacterial quantification

Nucleic acids were extracted from fecal stool pellets using the MagNA Pure 24 instrument and the MagNA Pure 24 Total NA Isolation Kit (Roche Diagnostics, Rotkreuz, CH) following repeated bead beating with Lysing Matrix Y beads (MP Biomedicals). *F. plautii* detection in fecal DNA extracts was performed with FP_F (TGAGTAACGCGTGAGGAACC) and PF_R (TCGTCGGGTACCGTCATTTG) primers[109]. Cloned PCR fragments amplified from F. plautii ATCC 6740 genomic DNA served as quantification standards. Quantitative PCR was conducted on a LightCycler 480 II instrument (Roche) using the LightCycler 480 SYBR Green I Master Kit (Roche) under the following conditions: 40 cycles (95 °C for 10 s, 60 °C for 15 s, 72 °C for 15 s) with an initial 10-min hot start at 95 °C. Total 16S rDNA copies were determined as previously described[110].

## Intestinal immune cell preparation

For intestinal immune cell preparations, ileum (defined as distal 10 cm of small intestine) was flushed with cold PBS, longitudinally opened to remove remaining content and cut into 1 cm pieces. After washing, the lamina propria dissociation kit (Miltenyi Biotec) was used according to manufactures instructions. Leucocytes were enriched on a 20/40/80% Percoll gradient (GE Healthcare Life Science, Pittsburgh, PA).

## Granulocyte infiltration assay

The granulocyte infiltration assay was performed as previously described[39]. In brief, ileal leucocytes were isolated as described above. Lamina propria cells were collected, washed with PBS and subsequently analyzed by flow cytometry.

## Crypt isolation

Crypts were isolated as described before[111]. In brief, large or small intestine were flushed with PBS, longitudinally opened to remove

remaining content and cut into 1 cm pieces. After washing until the supernatant remained clear, tissue pieces were incubated in PBS containing ethylenediamine-tetraacetic acid (EDTA) at 4 °C. Small intestine was incubated for 30 min in 5 mM EDTA, large intestine was incubated 3 times for 10 min with 30 mM EDTA. Small intestinal crypts were dissociated by forceful shaking 10-times. Large intestinal crypts were dissociated by 5-times forceful shaking for 30 s. Crypt-containing supernatant was filtered through one 100 µm and two 70 µm strainers or until the supernatant was clear.

For crypt isolation from human intestinal biopsies, the tissue pieces were washed twice in PBS and then digested with collagenase (Sigma Aldrich, C9697-50MG, 2 mg/ml in RPMI) for 30 min at 37 °C. Afterwards, the digestion solution was carefully removed, PBS added and the tissue disrupted by forceful pipetting. The crypt suspension was passed through a 100 µm strainer, resuspended in 50–100 µl of Matrigel and seeded into organoid cultures. The culture was started in human IntestiCult intestinal organoid growth media additionally supplemented with normocin (Invivogen, ant-nr-1, 100 µg/ml) and gentamicin (Gibco, 20 µg/ml) and then kept in WENR media.

### Organoid recovery assay
Intestinal crypt isolation was performed as described above. Time points of crypt isolation and used organs are depicted in the figure legends. After isolation, crypts were counted and a defined number ($n = 200$ crypts/drop for small intestine, $n = 300$ crypts/drop for large intestine) of crypts per drop used to initiate organoid cultures. At least two drops per mouse were seeded. Small intestinal organoids were cultured in ENR media. When small intestinal crypts were isolated from germfree or broadband antibiotic-treated mice, ENR media was supplemented with 10 µM Rock-inhibitor Y27632 to ensure stable organoid formation. Large intestinal organoids were cultured in IntestiCult intestinal organoid growth media supplemented with Rock-inhibitor Y27632. Established organoids were counted 3 days after seeding.

### Immunohistochemistry
After deparaffinization and re-hydration, slides were subjected to heat-induced antigen retrieval in a citrate-based buffer, pH 7.2 for 30 min using a microwave at 320 W. Blocking was done with peroxidase blocking solution, Dako REAL (cat no. S2023). Immunohistochemistry (IHC) was performed using the following primary antibodies: CD3 (1:50; clone SP7, catalog no. RM-9107-S1, Thermo Fisher), Ki67 (1:500, Dako, catalog no. M7249), cleaved Caspase-3 (1:1000 Cell Signalling, catalog no. 9661L). Primary antibodies were diluted in Antibody Diluent, Dako REAL (cat no. S2022) and incubated for 45 to 60 min at room temperature. Slides were washed using wash buffer from Dako/Agilent (10x concentrate, cat no. S3006). Secondary antibody (ready to use, cat no. 414341F, Histofine) was incubated for 45 to 60 min at room temperature. Detection was done using the DAB+ substrate detection system (catalog no. K3468, Dako). All stains were validated by internal and/or external positive controls as well as negative control specimens. The slide scanner Pannoramic 1000 Flash RX® (Sysmex Europe SE) was used for image acquisition, choosing the 20x objective. Virtual images from the slide scans were extracted for demonstration purposes using the software CaseViewer version 2.4 (Sysmex Europe SE).

The degree of CD3+ T-cell infiltration was examined in a blinded fashion by a board-certified pathologist. The degree of epithelial T cell infiltration was assessed by counting the number of infiltrating CD3+ cells (intraepithelial lymphocytes) in 10 high power fields (HPF) and calculating the average number of infiltrating cells per HPF for each specimen.

The GvHD pathology was examined in a blinded fashion by a board-certified pathologist (A.M. and D.H.). Histopathologic scores were assigned based on established criteria[112].

The number of cleaved Caspase-3+ cells was evaluated as the mean in 10 high-power fields.

The number of Ki67-positive cells was evaluated in individual, well-orientated crypts.

**Lgr5 in situ hybridization.** For detection of Lgr5 expression in FFPE mouse tissue, in situ hybridization (ISH) was performed using RNA-scope® 2.5 HD Reagent Kit-BROWN (322300; ACD, Hayward, CA, USA) with the RNAscope® Probe- Mm-Lgr5 (312171; ACD) specific for Lgr5 RNA in mouse. Tissue sections of 4 µm thickness were deparaffinized in xylene and ethanol and blocked with peroxidase (10 min). Slides were boiled in kit-provided antigen retrieval buffer at 95 °C for 15 min and digested afterwards with protease at 40 °C for 30 min. For hybridization, tissue sections were incubated with the target probe in the HybEZ hybridization oven (ACD) at 40 °C for 2 h. Pre-amplification and amplification steps were conducted using kit-provided reagents according to the manufacturer's recommendations. For signal detection, sections were incubated with the BROWN Reagent for 10 min at room temperature followed by counterstaining with hematoxylin, dehydration in ethanol and xylene and mounting with a xylene-based mounting medium.

**Mucus staining.** The mucus layer thickness was assessed as described before[113]. In brief, colonic sections containing stool pellets were collected 2 cm from the anus and transferred into fresh methanol-Carnoy's fixative (60% methanol, 30% chloroform, 10% acetic acid) for 24 h at room temperature. Tissue was then incubated twice with methanol for 30 min, twice in ethanol for 20 min, twice in xylene for 25 min and 1 h in paraffin. 5 µm-thick sections were used for mucus staining by Periodic acid-Schiff staining. To assess the thickness of the mucus layer, 8 evenly distributed sections were evaluated.

### In vitro T cell activation
For in vitro T cell assays, human pan T cells were isolated from PBMCs of healthy volunteers after informed consent (ethic vote 21-2224-101 Regensburg) with the human Pan T Cell Isolation Kit (Miltenyi Biotec) according to manufacturer's instructions. For stimulation, $2 \times 10^5$ T cells were seeded in a 96-well cell culture treated U-bottom well in T cell media and supplemented with 30 U/ml hIL-2 (Peprotec) and 2 µl of Dynabeads™ Human T-Activator CD3/CD28 beads (Thermo Fisher). Metabolites were added as indicated in the figure legends. After 24 h incubation at 37 °C and 5% CO$_2$, cells were harvested and stained for subsequent flow cytometry analysis.

### In vitro APC isolation
For in vitro activation of human APCs, CD14+ monocytes were isolated from PBMCs of healthy volunteers with the human CD14 microbeads (Miltenyi Biotec) according to manufacturer's instructions. CD14+ monocytes were either stimulated directly or differentiated into moDCs using the human Mo-DC Differentiation Medium (Miltenyi Biotec) according to manufacturer's instructions.

### In vitro organoid culture
Crypts were isolated as described above and suspended in liquefied growth factor reduced Matrigel (Corning) (33% ENR-medium or PBS; 66% growth factor-reduced Matrigel) at 4 °C, and were plated in delta-surface Nunc 24-well plates in 30 µl drops, each containing -200 crypts. All plates were incubated at 37 °C/5% CO$_2$ and ENR medium was replaced every 2–3 days. After 7 days organoids were passaged by mechanical disruption with a 5 ml syringe and an 18G needle.

For stimulation experiments, established organoids beginning after the first passage were used. In some experiments, we directly stimulated intestinal crypts after isolation without prior establishing time (referred as ex vivo crypt stimulation). To initiate stimulation experiments, organoids were passaged and seeded. Microbial metabolites were added after seeding (100 µM DAT or 50 µM ICA). In experiments using inhibitors, rapamycin (Invivogen, 10 nM–1 µM),

U0126 (Invivogen, 10 μM), chloroquine (1 μM) or H-151 (Invitrogen, 1 μg/ml) were added 4 h after seeding to prevent direct effects on the organoid formation after passaging. Metabolites were added 2 h after the inhibitors. For the evaluation of the MHC II expression on organoids, organoids were stimulated with metabolites for 3 days. Afterwards, organoids received additional stimulation with 2.5 ng/ml mIFNγ (Peprotec). Pictures for size determination were taken on day 6 of stimulation and analyzed in a blinded fashion using the ImageJ software. Flow cytometry analysis was performed on day 6 stimulation. On day 7, stimulated organoids were passaged. After 3 additional days, the number of established organoids after passage was assessed and normalized to the control group.

Human organoid cultures were performed with large and small intestinal organoids derived from tissue biopsies. Organoids were used from passage 3 onwards and cultured in WENR media that was replaced every other day. Otherwise, identical procedures as for murine organoids were used.

### T cell/organoid co-cultures without direct interaction

For murine co-cultures, T cells were isolated from splenocytes of donor BALB/c mice using the Pan T Cell Isolation Kit II, mouse (130-095-130, Miltenyi Biotec) according to the manufacturer's protocol. For co-culture, $1 \times 10^4$ allogeneic T cells were added to passaged C57BL/6 small intestinal organoids with murine IL-2 (30 μ/ml, Peprotec) and Dynabeads murine T-activator CD3/CD28 (ThermoFisher, 2 μl per well). To prevent direct contact between organoids and T cells, plates were kept slightly tilted with the organoid drop above the T cells placed at the bottom of the well. After 4 days, T cells were removed and media exchanged to ENR. Photos for size determination were taken at day 6 and area size of organoids was analyzed in a blinded fashion using the ImageJ software. Organoids were passaged after 7 days and counted on day 3 after passage. Before passaging, all organoid drops were microscopically inspected to exclude T cell infiltration into to the organoid drops.

For human co-cultures, a similar procedure was used. PBMCs from healthy human donors were isolated with Biocoll cell separation solution and T cells isolated with the human Pan T Cell Isolation Kit (Miltenyi Biotec) according to manufacturer's instructions. $1 \times 10^4$ allogeneic T cells were co-cultured with passaged organoids, human IL-2 (30 μ/ml, Peprotec) and Dynabeads human T-activator CD3/CD28 (ThermoFisher, 2 μl per well).

### In vitro intestinal GVHD model

The used in vitro model for intestinal GvHD was described in detail before[35]. In brief, small intestinal crypts of BALB/c mice were isolated and organoid cultures initiated as described above. For isolation of small intestinal intraepithelial lymphocytes (IEL), small intestine of C57BL/6 mice was removed, washed with PBS, cut into 0.5 cm pieces and twice incubated in HBSS (Sigma-Aldrich) + 10 mM HEPES + 5 mM EDTA + 5% FCS + 1 mM dithiothreitol (Sigma-Aldrich) for 20 min at 37 °C and for additional 20 min at 37 °C in HBSS + 10 mM HEPES. After each incubation step, the tissue pieces were vortexed and the supernatant containing IEL was collected. T cells were then enriched by a 40/70% Percoll gradient followed by a positive magnetic cell separation using anti-CD3-PE antibodies (Biolegend) and anti-PE MicroBeads, ultrapure (Miltenyi Biotec).

To set up the allogeneic co-culture, organoids were removed from Matrigel 2–4 days after crypt isolation. Approximately 100 organoids were incubated with $2.5 \times 10^5$ CD3+ magnetically enriched IEL T cells for 30 min at 37 °C. Afterwards, organoids and T cells were embedded in Matrigel and cultured in organoid media additionally containing 10 ng/ml mIL-7 (Immunotools), 10 ng/ml mIL-15 (Immunotools) and 100 μ/ml hIL-2 (Immunotools). Cell death in organoids was assessed after 2 days.

To assess cell death, organoids were stained with 10 μg/ml propidium iodide (Invitrogen) and 10 μg/ml Hoechst 33342 (Invitrogen)

for 30 min at 37 °C. Propidium iodide and Hoechst 33342 fluorescence were determined with the Tecan infinite M200 plate reader. To quantify cell death independently of the seeding density, intensity signal of the cell-impermeable dye propidium iodide as a surrogate for the amount of cell death was normalized to the fluorescence intensity of the cell-permeable dye Hoechst representing the number of cells in each well. Subsequently, to correct for physiological baseline cell death, the measured basal cell death, i.e., PI/Hoechst ratio was further normalized to the organoid control without T cells (i.e., normalized to BALB/c organoids cultured alone). Representative images of stained organoids were taken on a Leica DMI4000 B inverted microscope with a 10X objective.

### In vitro oxygen consumption

For oxygen consumption measurements of organoids were performed using the PreSens technology (PreSens Precision Sensing GmbH). Organoids were passaged and seeded above the sensor unit of a 24-well Oxodish OD24 plate and cultured for 3 days before measurement. 3 h after the last media change, measurements were started for the indicated time frames.

### Organoid irradiation

Small intestinal organoids were passaged and seeded as described above and stimulated as indicated. After 4 h, organoids were irradiated with 4 Gy. The number of viable organoids was assessed on d6 after irradiation. The number of regenerated organoids after passage was assessed 3 days after passage on day 7 after irradiation.

### Flow cytometry

For intracellular cytokine staining, cells were stimulated for 3–4 h with eBioscience™ Cell Stimulation Cocktail (plus protein transport inhibitors) (#00-4975-93). Cell suspensions were stained in PBS. Fluorochrome-coupled antibodies were purchased from eBioscience, BD or BioLegend. Data were acquired on a FACS LSR Fortessa X-20 or Symphony A5 (BD Biosciences) and analyzed using the FlowJo software (TreeStar). All staining protocols included 20 min staining with a live/dead marker and CD16/CD32 antibody (Biolegend) at 4 °C and surface marker staining for 30 min at 4 °C. For intracellular cytokine staining, the Foxp3 Transcription Factor Fixation/Permeabilization Kit (eBioscience) was used for fixation and permeabilization. Intracellular staining was performed for 60 min at 4 °C. All used antibodies and dilutions are summarized in Table S2.

### Cytokine measurements

Cytokine levels in cell culture supernatants or murine serum samples were determined using the LEGENDplex™ Mouse Inflammation Panel (13-plex) or the LEGENDplex™ Human Anti-Virus Response Panel (both Biolegend) according to manufacturer's instructions.

### Cell sorting

Single cell suspensions of small intestinal organoids were generated by removing organoids from the culture with PBS and digestion with TrypleE Express for 10 min at 37 °C. Single cells were passed through a 100 μM strainer, washed and stained with EpCAM-BUV395 (BD Bioscience) and respective hashing antibody (TotalSeq-B0301-306 anti-mouse Hashtag 1-6 Antibody, Biolegend). Live/dead staining was performed immediately before sorting by adding propidium iodide. Single cells were sorted as live/dead-EpCAM+ cells.

### Metabolite analysis

**Sample preparation for targeted analysis.** Approximately 100 mg of human stool was weighed in a 15 ml bead beater tube (Teen-PrepTMLysing Matrix D, MP Biomedical) and extracted with 5 ml of methanol. The samples were extracted with a bead beater (FastPrep-24™ 5G, MP Biomedicals) supplied with a CoolPrepTM cooling module

(MP Biomedicals, cooled with dry ice) 3 times each for 20 s with 30 s breaks in between a speed of 6 m/s. 1 ml of the fecal suspension was dried in a vacuum centrifuge (Eppendorf Vacufuge) to determine the dry weight.

## Measurement of SCFA, lactic acid, desaminotyrosine and indole-3-carboxyaldehyde

The 3-NPH method was used for the quantitation of desaminotyrosine and indole-3-carboxyaldehyde[114]. Briefly, 40 μl of the fecal extract and 15 μl of 50 μM isotopically labeled standards were mixed with 20 μl 120 mM EDC HCl·6% pyridine-solution and 20 μl of 200 mM 3-NPH HCL solution. After 30 min at 40 °C and shaking at 1000 rpm using an Eppendorf Thermomix (Eppendorf, Hamburg, Germany), 900 μl acetonitrile/water (50/50, v/v) was added. After centrifugation at 13,000 μ/min for 2 min the clear supernatant was used for analysis. The measurement was performed using a QTRAP 5500 triple quadrupole mass spectrometer (Sciex, Darmstadt, Germany) coupled to an ExionLC AD (Sciex, Darmstadt, Germany) ultrahigh performance liquid chromatography system. The electrospray voltage was −4500 V, curtain gas 35 psi, ion source gas 1 55 psi, ion source gas 2 65 psi and the temperature 500 °C. The MRM-parameters were optimized using commercially available standards. The chromatographic separation was performed on a 1.7 μm, 100 × 2.1 mm, 100 Å Kinetex C18 column (Phenomenex, Aschaffenburg, Germany) with 0.1% formic acid (eluent A) and 0.1% formic acid in acetonitrile (eluent B) as elution solvents. An injection volume of 1 μl and a flow rate of 0.4 mL/min was used. The gradient elution started at 23% B which was held for 3 min, afterward the concentration was increased to 30% B at 4 min, with another increase to 40% B at 6.5 min, at 7 min 100% B was used which was hold for 1 min, at 8.5 min the column was equilibrated at starting conditions. The column oven was set to 40°. To monitor instrument performance, every 25th sample was accompanied by the measurement of a quality control sample (calibration point 7), a derivatized solvent blank, and a solvent blank. Data acquisition and instrumental control were performed with Analyst 1.7 software (Sciex, Darmstadt, Germany). The data was analyzed with MultiQuant 3.0.3 (Sciex, Darmstadt, Germany) and Metaboanalyst[115]. Features with more than 70% of missing values were removed. Missing values were replaced by LoDs (1/5 of the minimum positive value of each variable). All measurements have been performed as a single measurement without technical replicates ($n = 1$).

## 16S rRNA gene amplicon sequencing

Murine fecal samples were collected at indicated time points during the experiment and snap frozen in liquid nitrogen. Samples were processed as previously described with some changes[116]. Briefly, after DNA isolation, targeted 16S rRNA gene amplicons were produced by fusing barcodes plus adapters using a 2-step PCR. Primers used have been previously described: 341F 5′-CCT ACG GGN GGC WGC AG-3′ and 785R 5′-GAC TAC HVG GGT ATC TAA TCC-3′[117]. For DNA isolation, the MaxWell RSC Faecal Microbiome DNA Kit (Promega) was used with bead beating. For the latter, the FastPrep-24 (MP Biomedicals Germany GmbH) was used, supplied with a CoolPrep adapter (MP Biomedicals, cooled with dry ice); thrice for 20 s of beating at a speed of 6 m/s, each followed by a 30 s break for cooling. Cycle times for the first and the second PCR had been adjusted for denaturation to 10 s, for annealing to 20 s and for extension to 45 s. After library preparation and equimolar pooling of the amplicons, the samples were sequenced using a MiSeq (Illumina) as described[116]. Raw sequencing data were analyzed in a UPARSE-based pipeline clustering sequences down to zOTUs (denoised operational taxonomic units) with www.imngs2.org[118], which includes the Taxonomy-Informed Clustering (TIC) algorithm[119]. Downstream analysis of zOTU tables was conducted using Rhea with default settings[120]. Briefly, reads were normalized[120] and spurious taxa were removed[121].

## Single cell RNA sequencing

**Sample preparation.** Murine small intestinal organoids from WT or STING Goldenticket mice were stimulated with DAT, ICA or carrier. Organoids were suspended to single cell suspensions as described above and labeled with hash-tag oligo (HTO) antibodies to enable pooling (cell hashing[122]). FACS was used to filter and sort a defined number of viable propidium iodide⁻EpCAM⁺ intestinal cells (Fig. S2A), different treatment groups were pooled in the process. Three experimental replicates with 6000 cells per treatment group were generated (36,000 cells in total per replicate).

**10x processing and sequencing.** Each replicate was split into two batches of 18,000 cells, and each batch was subjected to 10x Genomics processing to isolate cDNA of mRNA and HTOs of single cells using the Chromium Next GEM Single Cell 3′ Reagent Kits v3.1 and 3′ Feature Barcode Kit (Dual Index). The amplified libraries were then sequenced on an Illumina Nextseq 2000 device (P3 flow cell, all libraries multiplexed, 45% HTO libraries, 55% RNA-seq libraries) for initial probing. For two libraries, low average counts and a large degree of insufficient/ambiguous HTO labeling was observed. Consequently, only one (viable) library from each replicate was subjected to deeper sequencing on an Illumina Novaseq device (S2 flow cell, all libraries multiplexed, 12.5% HTO libraries, 87.5% RNA-seq libraries). All sequencing runs were performed using the following parameters: PE-28-10-10-90. Reads from both sequencing runs were combined in the subsequent analysis.

**Raw read processing.** Base-calling and demultiplexing of sequencing reads was carried out using bcl2convert provided by Illumina Dragon (v-3.8.4, NextSeq2000) and DragonServer (v2.1, NextSeq 6000) (Illumina Inc., San Diego, California, USA). Sequencing reads were combined and mapped to the mouse genome (https://cf.10xgenomics.com/supp/cell-exp/refdata-gex-mm10-2020-A.tar.gz, based on Mus-musculus.GRCm38.dna.primary_assembly.fa.modified and gencode.vM23.primary_assembly.annotation.gtf.filtered) using Cellranger (v 7.0.0) with standard options and expected cells set to 10000. Count files for gene expression libraries and Cell-surface-marker (HTO) libraries were created by Cellranger.

**Count data preprocessing and quality control.** Processing and analysis of single cell RNA-seq and HTO count data was mainly performed in R 4.1.1 with the *Seurat* package v4.0.5. Since cell hashing was applied, it was possible to pool different treatment groups in a single RNA-seq library. In order to assign treatment groups to single cells based on HTO counts, the *HTODemux()* function from the *Seurat* package was employed with default settings (positive-quantile parameter of 0.99) after centralized log normalization. Cell calling via *Cellranger* yielded a total of 48,797 cells across the three libraries and all conditions. Cells that could not be assigned a treatment group due to insufficient HTO labeling (negative cells) were excluded from the analysis, resulting in 22,082 remaining cells. The large proportion of negative cells can likely be explained by a marked fraction of lysis and the deep sequencing, which makes it more likely that cell fragments are called as cells.

Doublets were identified based on ambiguous HTO labeling reported by *HTODemux()* and using the *scDblFinder* R-package[123] (version 1.8.0, used together with *SingleCellExperiment*, version 1.16.0) and also excluded from the analysis. *scDblFinder* was run in cluster-based mode using the top 30 principal components (PCs) associated with the 3000 top expressed genes to build the k-nearest-neighbor (kNN) network. The top 15 principal components were included when training the gradient boosting classifier. Three iterations of scoring were performed. Known doublets from cell hashing were supplied, but only used for score thresholding and not for training. To obtain clusters for use with *scDblFinder*, the *FindNeighbours()* and *FindClusters()* functions of *Seurat* were used with default parameters on each scRNA-

seq library separately after excluding negative cells, using the top 30 principal components as input. The clustering resolution was 0.5. Clustering and data preparation for clustering are described in more detail in the following section. To obtain an estimate for the expected doublet rate to supply to *scDblFinder*, the probability $p_g$ that a doublet is formed from the same treatment group, as well as the probability $p_c$ that a doublet is formed from the same cluster (homotypic doublet) were estimated based on the respective group and cluster sizes. Then the expected doublet rate $r_e$ is estimated as $r_e = \frac{r_h \cdot (1 - p_c)}{1 - p_g}$, where $r_h$ is the observed doublet rate from cell hashing. However, *scDblFinder* is only very loosely bound to the $r_e$ values supplied. After removing doublets, 16,629 cells remained.

In order to obtain a valid set of cells for analysis, cells have additionally been filtered to have a total RNA count between 17,500 and 130,000, a total number of features between 3000 and 10,000 and a maximum percentage of mitochondrial genes of 7.5%. In this step, filtering removed about 45%, 55% and 80% of cells for the three libraries, respectively. For the library with 80% removal, the reason was a large fraction of (non-negative) cells with low counts and high mitochondrial gene percentage, indicating lysis. Across all libraries, a total of 6229 cells were then used for subsequent analyzes.

**Dimensionality reduction and clustering.** Counts were normalized for library size by dividing by the sum of total counts per cell and multiplying with a scale factor of $1 \cdot 10^6$. Afterwards, a value of 1 was added and the natural logarithm of the counts was computed. The 3000 top variable features were identified by applying *Seurat's FindVariableFeatures()* function with default parameters (variance stabilizing transformation as selection method). Data was then mean-centered, scaled to unit variance, and subjected to principal component analysis (PCA). The principal components (PCs) were then used as input to the *RunHarmony()* function from the *harmony* R-package[124] (version 0.1.0) to integrate the data from the different libraries by removing batch effects. *RunHarmony()* was employed with default parameters, grouping by library.

Afterwards, *Seurat's FindNeighbours()* function was used with the first 30 dimensions of the *harmony*-transformed data as input to construct a nearest-neighbor graph. The *FindClusters()* function was then run with default parameters (standard Louvain algorithm) to perform graph-based clustering on the cells. Clustering was run at different resolutions to find the optimal clustering setup. For visualization, a UMAP (Uniform Manifold Approximation and Projection[125]) dimensionality reduction was additionally applied to the data, using *Seurat's RunUMAP()* function on the first 30 dimensions of the *harmony*-transformed data. A resolution of 0.8 was chosen for the final clustering setup, as it provided the best compromise between granularity and meaningfulness of clusters, as well as agreement with the localization of cells in UMAP space. There were no pronounced differences in clustering or UMAP projection across libraries.

**Cell annotation.** For cell type annotation of single cells, the R-package *SingleR* (version 1.8.1)[126] was used, together with a scRNA-seq reference dataset from the murine intestine from Haber et al. (2017, GEO accession number GSE92332, full length atlas data)[46]. The reference labels taken from that data set were ISCs, TA (transient amplifying) cells, enterocyte progenitors (late and early were combined), enterocytes, enteroendocrine, goblet, paneth and tuft cells. As the method for determining reference label marker genes, the Wilcoxon rank-sum test was used, otherwise *SingleR* was run with default parameters. Briefly, *SingleR* computes correlations between cells from the query dataset and annotated cells from a reference dataset, based on a set of reference label marker genes inferred from the reference dataset. A given cell is then annotated with the reference label it correlates best with. To validate the *SingleR*-based annotation, scores of cell type marker gene signatures have additionally been computed for each cell,

using the *UCell* R-package (version 2.0.1, together with R version 4.2.0)[127]. To this end, marker gene signatures for ISCs, enterocytes, goblet, paneth, enteroendocrine and tuft cells as well as cell cycle marker genes have been taken from Haber et al.[46], where they are shown in Extended Data Fig. 1. Additionally, cell cycle marker gene signatures specific to either G2M- or S-phase have been taken from *Seurat*, they are originally from Tirosh et al.[128]. *UCell* provides robust signature scores based on the Mann-Whitney U statistic[127]. It has been run with a maximum of 3000 ranked genes per cell, otherwise default parameters were employed. UMAP plots of single cells, colored by *UCell* scores of the given cell type signature, could then be inspected to derive possible cell type annotations of sets of cells, and be checked for agreement with the *SingleR*-based annotation.

When aligning *SingleR*-based annotation with clustering results in UMAP space, which can be seen in Fig. S8A and B, respectively, enterocyte and enterocyte progenitor, as well as goblet/paneth cell and enteroendocrine cell annotation is in acceptable agreement with the obtained clusters. Neither clustering nor annotation could distinguish goblet and paneth cells, though, and only very few cells could be confidently annotated as tuft cells. Also, the clustering does not reflect the distinction between ISCs and TA cells, which can be seen as a trend from left to right (lower UMAP 1 to higher UMAP 1). This distinction can also not be properly reproduced in clustering when increasing the resolution. However, when considering Fig. S8B, marked clustering of the large center population in UMAP 2 direction can be observed, which is not reflected by cell type annotation. Based on the UMAP plots of cell cycle signature *UCell* scores in Fig. S9A–D, we assume that this clustering results from differences in cell cycle stage, where the upper-center population shows high G2M-phase signatures, while the lower-center population shows high S-phase signatures.

Another reason why the clustering does not fully align with cell type annotation is that the transition from ISCs over TA to enterocyte progenitor cells would be expected to be smooth, with probably rather subtle differences. This is supported by the heatmap of centered and scaled marker gene expression in Fig. S10A. The columns correspond to single cells with *SingleR* celltype annotation, while the rows correspond to the expression of a given marker gene from a reference signature. While ISCs cells show a higher expression of stem cell markers than TA cells, their overall profile is relatively similar. A progression from ISCs cells over TA cells and enterocyte progenitors to enterocytes is perceivable. This transition is also visible in the heatmap of *SingleR* scores in Fig. S10B. The columns again correspond to single cells with *SingleR* celltype annotation, while the rows correspond to the *SingleR* scores for a given cell type label from the reference. Higher scores mean a higher correlation with and thus similarity to the reference label. *UCell* scores of marker gene signatures support the *SingleR* annotation. This can be seen in the UMAP plots of *UCell* scores in Fig. S9A–H: The stem cell score is highest in the leftmost part of the large center population. It then diminishes from left to right, indicating the aforementioned smooth transition. However, the smoothness of the transition also means that it is hard to define strict boundaries for ISCs and TA cells, and that the annotation is fuzzy/approximate. The enterocyte score is very high in the bottom population of the UMAP projection. It then diminishes upwards, indicating enterocyte progenitor cells. This matches the enterocyte and enterocyte progenitor annotation from *SingleR*. Of course, agreement between signature *UCell* scores and *SingleR* annotation is not unexpected, as the annotation of the scRNA-seq reference dataset from Haber et al.[46] was done with these gene signatures. Nevertheless, provided the correctness of the reference dataset annotation, the match supports the validity of the *SingleR* annotation.

**Trajectory-based cell annotation.** As previously mentioned, the distinction between ISCs and TA cells in the *SingleR* annotation was fuzzy and not captured by the clustering. Thus, using the R package

*Monocle*[129] (version 2.22.0), a trajectory analysis of the subpopulation comprising ISCs and TA cells was performed, with the aim of being able to better represent the smooth transition between cell states. To this end, marker genes that distinguish ISCs and TA cells in the Haber et al. (2017) reference dataset were searched for using *Seurat*'s *FindMarkers()* function on the log-normalized reference dataset (normalization procedure as described above for the dataset from this study). *FindMarkers()* was run with the Wilcoxon Rank Sum test to identify differentially expressed genes, excluding only genes that were expressed in less than 10% of cells in both groups. Differential expression testing results were then filtered for a Bonferroni-adjusted $p$-value ≤ 0.05 and an absolute $\log_2$ fold change ≥0.25 to obtain the list of marker genes. Based on this set of genes, *Monocle* was then used to construct the trajectory with the *DDRTree* dimensionality reduction method. When constructing the trajectory, the *residualModelFormulaStr* parameter was used to regress out effects explained by library, experimental condition, as well as the S-phase and G2M-phase UCell scores (to avoid distortion of the trajectory due to cell cycle effects). Figure S11A shows plots of the resulting trajectory colored by various features relating to stemness, trajectory-pseudotime and annotation. The expression of the stem cell markers *Lgr5* and *Olfm4* as well as the *UCell* stem cell score increase with *DDRTree* Component 1, while the trajectory-derived pseudotime decreases with Component 1. Thus, stemness decreases with pseudotime, and pseudotime reflects the transition from ISCs to TA cells. When coloring the trajectory with the *SingleR* based annotation, TA cells are found mostly at low Component 1, while ISCs are found at high values of Component 1. At intermediate values, a mixture of the two cell types can be observed. Based on pseudotime, stemness markers and *SingleR* annotation, we thus annotated the trajectory-identified state with high Component 1 (prior to the first branching point) as ISCs and late states with low Component 1 as TA cells. The intermediate states were annotated as ISC II, reflecting an intermediate population that exhibits a mixture of ISC and TA characteristics.

In addition to the stemness progression seen along Component 1, a branching of the trajectory can additionally be observed in the direction of Component 2 for ISC II and TA cells. When considering the trajectory plots with G2M- and S-phase cell cycle *UCell* scores in Fig. S11B, there does not seem to be any cell cycle-related trend leading to the separation of cell populations in Component 2. Given that we regressed out cell cycle effects when constructing the trajectory, we indeed would not expect pronounced cell-cycle-related patterns. To be able to better assess what drives the trajectory separation in Component 2, we created an additional annotation with multiple subsets for ISC II and TA cells based on trajectory states along Component 2 (see Fig. S11B, right panel). We then compared populations with higher Component 2 to populations with lower Component 2 using the Wilcox-Test with subsequent GSEA analysis. Overall, we found that low-Component-2 populations exhibit an upregulation especially of ribosome-, translation- and energy-production-related gene sets, which may indicate cellular biomass production for growth and proliferation. High-Component-2 populations, on the other hand, show a noticable upregulation of Rho-GTPase associated gene sets and the KEGG adherens junction gene set. As Rho-GTPases are key regulators of cytoskeletal rearrangement[130], this could for example be indicative of cell migration, cell cycle associated rearrangements (while such effects should not be too pronounced, given that we attempted to remove cell cycle related effects during trajectory construction) or cell polarity changes. A possible explanation for these observations could thus be the migration of cells to the appropriate organoid position in the process of differentiation, probably in association with changes in cell polarization and adhesion. However, given that our focus was on cellular characteristics that vary with the stemness progression along trajectory Component 1, we did not perform a more in-depth investigation of the cell populations along Component

2. To independently assess the trajectory-based categorization of stem-like cells into ISC, ISC II and TA also in UMAP space, a UMAP representation of the ISC/TA subset of cells was computed. To this end, the subset of cells was processed with *Seurat*, mean centering and scaling the data to unit variance, and performing PCA on the set of marker genes also used for trajectory analysis. Subsequent processing of the PCA data, including *harmony* integration, was done as described for the full dataset, followed by UMAP transformation. The resulting UMAP representation can be seen in Fig. S11C colored by various features relating to stemness, trajectory-pseudotime and annotation, as for the trajectory plots. The shape of the point cloud in UMAP space is rather undefined, indicating relatively small differences in the analyzed subset of cells. However, the stemness and pseudotime trends, as well as the separation of the ISC, ISC II and TA entities (as annotated according to trajectory states) can also be observed in UMAP space, independently from trajectory construction.

**Differential gene expression and gene set enrichment analysis.** For each population of cells annotated with a given cell type (except tuft cells, due to the very low number of cells), testing for differential gene expression between treatment groups was performed on the raw count data with the *NEBULA* R-package[131] (version 1.2.2). *NEBULA* fits (by default) a negative binomial mixed model to the count data, and estimates both cell- and subject-level overdispersions. For each cell type, a *NEBULA* negative binomial gamma mixed model was fitted with treatment group and library (batch) as categorical variables, as well as the number of features per cell as a continuous variable (to account for possible effects not captured by a library size scaling factor). Each combination of treatment group and experimental replicate was considered a distinct subject. Library size was set as the scaling factor, and minimum counts per cell for a gene to be tested were set to 0.005. The fit was computed with the *NEBULA*-LN method. All other parameters were set to default values. In order to be able to make all desired comparisons, contrasts were employed. To that end, the covariance matrix of the model was extracted and used together with the log fold-change (log FC) estimates and a vector of linear contrasts in a chi-squared test, as described in the vignette of the *NEBULA* package. When testing for differential gene expression between cell types/states, irrespective of treatment group, the Wilcoxon Rank Sum test from *Seurat*'s *FindMarkers()* function was used on the log-normalized data, employing Bonferroni correction for multiple testing. To be as unrestrictive as possible, all genes expressed in at least 1% of cells in either of the two compared cell types/states were included in testing.

For pathway/gene set analysis, the GSEA (gene set enrichment analysis)[132] implementation from the *fgsea* R-package (version 1.20.0) was employed. As input for GSEA, the negative $\log_{10}$ $p$-values signed with the direction of the log FC were used. If there were $p$-values with a value of zero, these were set to the smallest non-zero normalized floating-point number ($2.23 \cdot 10^{-308}$) prior to log-transformation. Values were then sorted by the signed log-transformed $p$-values in descending order, breaking possible ties by sorting according to log FC. Gene sets were obtained from MSigDB (Molecular signatures database)[133] via the *msigdbr* R-package (version 7.4.1). Chosen gene set collections were hallmark gene sets (H), the KEGG and Reactome subsets from curated gene sets (C2), as well as transcription factor targets from regulatory target gene sets (C3). Only gene sets where at least 70% of genes were in the genes tested for differential expression in the given comparison were assessed. The $p$-values obtained from GSEA for the different comparisons and gene sets were corrected for multiple testing with the false discovery rate (FDR) approach[134], controlling for an FDR of 10%. For comparisons between treatment groups within a given cell type, correction was performed across all hypotheses associated with comparisons in that cell type. For comparisons between cell types/states, correction was performed across all hypotheses associated with a given comparison of cell types/states.

**Cell type abundance analysis.** Analysis for differential cell type abundance (DA) between treatment groups was performed with the *glm.nb* function from the *MASS* R-package[135] (version 7.3.54). Briefly, for each cell type from the annotated scRNA-seq data (except tuft cells), a negative binomial model with the variables treatment group and library (batch) was fitted to the cell count data. Total cell count per sample (i.e., library size) was used as an offset for normalization. *P*-values were derived by performing a Wald test with the respective linear contrasts of model parameter estimates. They were then adjusted for multiple testing using the FDR approach across all hypotheses associated with comparisons within a given cell type, controlling for an FDR of 10%.

**Additional software/packages.** Other R-packages used were *ggplot2* (version 3.3.5), *patchwork* (version 1.1.1), *dplyr* (version 1.0.7), *stringr* (version 1.4.0) and *tidyr* (version 1.1.4), *pheatmap* (version 1.0.12) and *viridis* (version 0.6.2), as well as *foreach* (version 1.5.1), *doParallel* (version 1.0.16), *future* (version 1.31.0), *rprojroot* (version 2.0.2), *yaml* (version 2.2.1), and *WriteXLS* (version 6.3.0).

## Statistics

All statistical tests used are named in the corresponding figure legends. If not depicted otherwise, all data is shown as mean ± SEM. *P*-values are shown in the graphs.

In general, data were checked for normality by Shapiro–Wilk test. For pairwise comparisons, an unpaired *t*-test was used for normally distributed data. Otherwise, a Mann–Whitney U test was applied. For comparisons of three or more different groups, ANOVA with Tukey's multiple comparison test was used. When data did not meet the ANOVA criteria, a Kruskal–Wallis test with Dunn's post-hoc test was utilized. Normalized data was analyzed by one sample *t*-test. The null hypothesis was then defined as the mean of a treatment group to equal 1. The null hypothesis was rejected in the case of (adjusted) $p \leq 0.05$. Differences in survival were analyzed by log-rank (Mantel–Cox) test. All in vitro data were validated in at least three independent experiments. All one- and two-sample tests were run two-sided. Graphpad Prism V10 was used for statistical analysis if not stated otherwise.

## Reporting summary

Further information on research design is available in the Nature Portfolio Reporting Summary linked to this article.

## Data availability

Single cell RNA-seq data of murine organoids were deposited at Gene Expression Omnibus (GEO) (accession number: GSE261714, https://www.ncbi.nlm.nih.gov/geo/query/acc.cgi?acc=GSE261714). The publicly available dataset we utilized for cell type annotation in our single cell RNA-seq data of murine organoids is available at Gene Expression Omnibus (GEO) (accession number: GSE92332, https://www.ncbi.nlm.nih.gov/geo/query/acc.cgi?acc=GSE92332). Primary mass spectrometry data from patient stool samples have been annotated with clinical metadata and deposited at Zenodo under accession number 6603017. 16S data were deposited at the European Nucleotide Archive (ENA) (accession number; ERP180829, https://www.ebi.ac.uk/ena/browser/view/ERP180829). The remaining data are available within the Article, Supplementary Information or Source Data file. Any additional information is available from the corresponding authors. Source data are provided with this paper.

## Code availability

The scripts used for scRNA-seq analysis have been deposited at GitHub (https://github.com/lit-regensburg/scRNA_intest_org_metabolites; https://doi.org/10.5281/zenodo.16847406).

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

## Acknowledgements

This study was supported by Deutsche Forschungsgemeinschaft – Projektnummer 360372040 – SFB 1335 (to H.P., S.H., F.B. and J.R.), Projektnummer 395357507 – SFB 1371 (to H.P., E.T.O., E.H., E.M., K.N., K.P.J., J.R., K.K, M.T., D.H., A.G.), Projektnummer 324392634 – TRR 221 (to H.P., E.H., M.B-H., M. K., M.E., D.W., D.W., M.R., C.S., P.H., and W.H., A.G.), Projektnummer 375876048- TRR241 (to. K.H.), BA 2851/6-1 and BA 2851/7-1 to F.B., the German Cancer Aid (70114547 to H.P.), Projektnummer 441891347 – SFB 1479 (to N.K.), DFG under Germany's Excellence Strategy, CIBSS – EXC 2189 (Projektnummer 390939984, to N.K.), the Wilhelm Sander Foundation (2021.040.1 to H.P.), the Bavarian Cancer Research Center (BZKF) (to. M.P. and H.P.), The Else- Kröner-Fresenius-Stiftung (funding line: Else-Kröner Forschungskolleg to E.T.O. and E.M.), the Deutsches Konsortium für Translationale Krebsforschung (fellowship to E.T.O.). The Bavarian State Ministry of Science and Art (COVID funds to H.P.), the ReforM program of the University Regensburg (to M.P.). E.T.O. is supported by the DGIM Clinician Scientist Program. H.P. is supported by the EMBO Young Investigator Program. This work was funded/Co-funded by the European Union (project MICROBOTS, Grant No. 101124680 to H.P.). Views and opinions expressed are however those of the author(s) only and do not necessarily reflect those of the European Union or the European Research Council. Neither the European Union nor the granting authority can be held responsible for them. This work was performed in the framework of the National Cancer Therapy Center (NCT) WERA. This work is part of the doctoral thesis of S.G. at the Technical University of Munich. We thank Tatiana Nedelko, Maria Krieger (Heidegger and Poeck group) and Doris Gaag (Pathology UKR) for excellent technical support. Single-cell RNA sequencing was performed at the NGS Core of the LIT (Regensburg, Germany). We thank Hanna Stanewsky for technical support. Germfree mouse experiments were performed at the germfree mouse facility of UKR (Regensburg, Germany). We thank Nico Kumher and Karina Käufl for technical support. FACS analyzes were performed at the FACS core of the LIT (Regensburg, Germany). Schematics were generated with Biorender.

## Author contributions

Conceptualization: S.G., E.H., H.P. Data curation: S.G., E.T.O., P.H. Formal Analysis: S.G., E.T.O., K.F., P.H. Investigation: S.G., E.T.O., K.F., P.H., D.M.M., O.K., L.K., C.Su., N.S., C.G., E.V., A.M., D.H., E.M., A.H., S.Gh., L.J., F.V., F.P. Methodology: S.G., K.F. P.H., K.K., J.G., K.N., K.P.J., J.R., M.K., C.Sc., N.K., M.T. Project administration: S.G., H.P. Resources: K.K., K.N., J.R., M.K., D.W., C.Sc., N.K., P.Ho., M.E., D.Wo., F.B., M.R., D.H., M.E., K.H., M.B.H., W.H., A.G., S.H., E.H., H.P. Supervision: S.G., E.T.O., E.H., H.P. Visualization: S.G., E.T.O., P.H., E.M., A.H., M.P. Writing – original draft: S.G. Writing – review & editing: all authors Funding acquisition: C.Sc., S.H., E.H., H.P. Contributions are specified according to CRediT (Contributor Roles Taxonomy). All authors reviewed and approved the final version of the manuscript.

## Funding

## Competing interests

HP: honoraria: Novartis, Gilead, Abbvie, BMS; Pfizer, Servier; Janssen-Cilag travel: Gilead, Janssen-Cilag, Novartis, Abbvie, Novartis; Jazz, Amgen Research: BMS. DW received research support from Novartis and honoraria from Sanofi, Incyte, Behring, Mallickrodt, Neovii and Takeda. F.B.: honoraria: BMS, Janssen, travel: Janssen. M.B-H.: honoraria: Pfizer Pharma GmbH; Sanofi Deutschland GmbH; travel: Novartis

Pharma GmbH. S.H. is an employee of and hold equity interest in Roche/Genentech. WH: honoraria: Amgen, Novartis; travel: Janssen-Cilag, Amgen. E.T.O.: honoraria: BeiGene, AstraZeneca, travel: Janssen, Lilly. The content of this manuscript is part of a patent application of S.G., E.T.O. and H.P. All remaining authors declare no competing interest.

## Additional information

Sascha Göttert[1,2,22], Erik Thiele Orberg [1,22] ✉, Kaiji Fan[1,2], Paul Heinrich [1,2], Diana M. Matthe[3,4], Omer Khalid[1,2], Lena Klostermeier[1], Chiara Suriano[1,2], Nicholas Strieder [2], Claudia Gebhard[2], Eva Vonbrunn [5], Andreas Mamilos[6,7], Daniela Hirsch[6], Elisabeth Meedt[1], Karin Kleigrewe [8], Andreas Hiergeist[9], Alix Schwarz[10], Joachim Gläsner[9], Sakhila Ghimire[1], Laura Joachim[1], Florian Voll [1], Klaus Neuhaus [11], Klaus-Peter Janssen [12], Markus Perl[1], Franziska Pielmeier[1,2], Jürgen Ruland [13,14,15], Marina Kreutz[1], Daniela Weber[6], Christian Schmidl [2], Natalie Köhler [16,17], Markus Tschurtschenthaler [14,18,19], Petra Hoffmann [1,2], Matthias Edinger[1,2], Daniel Wolff[1], Florian Bassermann [10,13,14,20], Michael Rehli[1,2], Dirk Haller [11,21], Matthias Evert[6], Kai Hildner[3,4], Maike Büttner-Herold[5], Wolfgang Herr[1], André Gessner[9], Simon Heidegger [10,14], Ernst Holler[1,23] & Hendrik Poeck [1,2,20,23] ✉

[1]Department of Internal Medicine III, Hematology & Internal Oncology, University Hospital Regensburg, Regensburg, Germany. [2]Leibniz Institute for Immunotherapy (LIT), Regensburg, Germany. [3]Department of Medicine 1, Kussmaul Campus for Medical Research, University Hospital Erlangen, University of Erlangen-Nuremberg, Erlangen, Germany. [4]Deutsches Zentrum Immuntherapie (DZI), University Hospital Erlangen, Erlangen, Germany. [5]Department of Nephropathology, Institute of Pathology, Universitätsklinikum Erlangenm, Friedrich-Alexander-University Erlangen-Nürnberg (FAU), Erlangen, Germany. [6]Institut of Pathology, University of Regensburg, Regensburg, Germany. [7]Department of Pathology, German Oncology Center, Limassol, Cyprus. [8]Bavarian Center for Biomolecular Mass Spectrometry (BayBioMS), Technical University of Munich, Freising, Germany. [9]Institute of Clinical Microbiology and Hygiene, University Hospital Regensburg, Regensburg, Germany. [10]Department of Internal Medicine III, School of Medicine and Health, Technical University of Munich, Klinikum rechts der Isar, Munich, Germany. [11]ZIEL - Institute for Food & Health-, Technische Universität München, Freising, Germany. [12]Department of Surgery, TUM School of Medicine and Health, Klinikum rechts der Isar, Technische Universität München, Munich, Germany. [13]German Cancer Consortium (DKTK), partner-site Munich, a partnership between DKFZ and Klinikum rechts der Isar, Munich, Germany. [14]Center for Translational Cancer Research (TranslaTUM), School of Medicine and Health, Technical University of Munich, Munich, Germany. [15]Institute of Clinical Chemistry and Pathobiochemistry, School of Medicine and Health, Technical University of Munich, Munich, Germany. [16]Department of Internal Medicine I, Hematology, Oncology, and Stem Cell Transplantation, Faculty of Medicine, Medical Centre, University of Freiburg, Freiburg, Germany. [17]Centre for Integrative Biological Signalling Studies (CIBSS), University of Freiburg, Freiburg, Germany. [18]Division of Translational Cancer Research, German Cancer Research Center (DKFZ) and German Cancer Consortium (DKTK), Heidelberg, Germany. [19]Chair of Translational Cancer Research and Institute of Experimental Cancer Therapy, TUM School of Medicine and Health, Klinikum rechts der Isar at Technical University of Munich, Munich, Germany. [20]Bavarian Cancer Research Center (BZKF), Munich, Germany. [21]Chair of Nutrition and Immunology, Technische Universität München, Freising, Germany. [22]These authors contributed equally: Sascha Göttert, Erik Thiele Orberg. [23]These authors jointly supervised this work: Ernst Holler, Hendrik Poeck. ✉e-mail: erik.orberg@ukr.de; hendrik.poeck@ukr.de

