## [Transparent Peer Review file · Nature Communications]

The microbial metabolite desaminotyrosine protects against graft-versus-host disease via mTORC1 and STING-dependent intestinal regeneration

Corresponding Author: Dr Erik Thiele Orberg

Version 0:

Reviewer comments:

Reviewer #1

(Remarks to the Author)

In the manuscript titled "A microbial metabolite protects against graft-versus-host disease via mTORC1 and STING-dependent intestinal regeneration" by Göttert et al., the authors have investigated the role of the microbial metabolite DAT in preventing GvHD in the context of allo-HSCT for leukemia therapy. The authors have demonstrated that DAT not only prevents GvHD but also promotes GvL responses. They have provided clear evidence that DAT promotes intestinal stem cell proliferation through mTORC1- and STING-dependent pathways, independent of type-I interferon receptor. However, the mechanism by which DAT modulates graft-versus-leukemia responses remains less clear. While this discovery is intriguing, there are some concerns that need further clarification.

Major Comments:

- 1) Did the number of crypt-like budding structures per spheroid also increase? If so, the role of DAT in intestinal stem cell (ISC) differentiation should also be investigated. Moreover, for consistency, the effect of DAT on ISC proliferation and irradiation-induced cell death should be examined *in vivo*.
- 2) Did the meropenem plus DAT treatment or *F. plautii* transfer alone actually improve GvHD and GvL *in vivo*? Additionally, does Sting-deficiency affect DAT's protective effect on GvHD and GvL *in vivo*?
- 3) The effects of DAT on Tregs and CD8 T cells appear inconsistent between the *in vitro* and *in vivo* data, especially at low doses of DAT. Is it possible that the promoting effect of DAT on GvL is through innate immune activation, considering that DAT administration was stopped before allo-T cell transfer?

Minor Comments:

- 1) Why was DAT administered only for 7 days before allo-BMT (Fig. 1D) or 9 days before allo-T cell transfer (Fig. 1F), while ICA was given continuously before and after allo-BM and allo-T cell transfer? If DAT is given at a relatively low dose continuously, would its protective effect disappear? How rapidly do serum levels of DAT drop after discontinuing administration?
- 2) Typo errors: For example, in Fig. 1C and I, "probability of survival" should be "percentage of survival"; in Fig. 1J, "probability" should be "relapse rate"; in the main text L457, "Fig 6B" should be "Fig 6C".

(Remarks on code availability)

Reviewer #2

(Remarks to the Author)

This is an interesting study to show that the levels of microbial product desaminotyrosine (DAT) are associated with better outcome in allo-HSCT patients. Pretreatment of mice with DAT or DAT producing probiotics improved survival and intestinal regeneration of allo-HSCT after total body irradiation (TBI). The benefit of DAT was abrogated in germ-free or STING-deficient (KO, Gt/Gt) mice. The GI protective effects in vivo (survival, histology, inflammation and Lgr5 preservation) are supported by strong data. In organoids, DAT promotes growth of untreated culture as well and that under many conditions, i.e, activated T cells, radiation, chemo, with or without EGF, germ-free or antibiotic treatment. STING KO but not IFNAR1 KO organoids showed loss of DAT induced growth. The author previously published that DAT promotes T cell-dependent anticancer immunity. In this study, the contribution from T cell activation or immune modulation seems minor but perhaps not surprising as the compound was given before TBI and allograft. Overall, the approach is comprehensive, and the current data support a cell intrinsic protective mechanism. However, the mechanistic studies missed a critical aspect on injury protection/reduction as detailed below.

- 1) DAT was given for 7 days before TBI and allograft. Even high-level DAT pretreatment was trending protection (Fig. 2 and 7C). Is continuous treatment of low dose also detrimental? Fig. 5, the growth promoting phenotype in organoids needs confirmation in mice by accessing proliferation, mTOR/metabolic signaling, and Lgr5 cell expansion in WT and STING KO mice after 7 day of DAT treatment, but before TBI and allograft (day 0). Day 7 is outcome.
- 2) Fig. S2C, 2G indicated that DAT pretreatment resulted in radio and chemo resistance in organoids, and the barrier effects in mice were pronounced on day 3. TBI causes acute crypt damage through p53-PUMA dependent loss of Lgr5+ stem and TA cells, which is exacerbated likely by allograft T cells. Activation of the PI3K/AKT and Wnt pathways (growth factors, Rspodin, GSK inhibitors) blunts Lgr5 and crypt apoptosis and improves intestine regeneration, only when given prior to exposure. Fig. 5G. does DAT treatment affect Lgr5+ cell numbers, crypt apoptosis/death (TUNEL) on 0, 1 and 3, and any effects on the p53 pathway or targets?
- 3) Fig. 4 In organoids, STING-dependent growth promoting phenotype is novel and interesting, but cannot be directly extrapolated to the in vivo injury model. mTORi or antioxidant is not STING specific. It is difficult to understand "deregulated mTOR signaling" or metabolism by STING KO based on pathway analysis alone. Inclusion of relevant DEG tables can be helpful. Radiation should be used to better address cell intrinsic mechanism. Does STING KO abrogated DAT-mediated radioprotection, how about the effects on mTOR signaling/metabolism, apoptosis, and the p53 pathway?
- 4) Fig 8. The immune compartment and the ISC compartment seem disconnected in "injury protection" model.
- 5) Discussion. "Type 1 IFN-independent". Prior studies established that epithelial intrinsic STING and IFN beta is required for intestinal regeneration after TBI (PMID: 28424327; PMID: 34613772). Either loss of DAT-induced Lgr5 expansion before injury or failed regeneration after can lead to similar outcomes in STING KO mice. This point requires discussion.

(Remarks on code availability)

scRNA-seq data has been deposited.

Reviewer #3

(Remarks to the Author)

Gottert & coauthors studied the role of desaminotyrosine (DAT), a microbiome-derived metabolite, in GVHD. This comprehensive study made use of observational cohorts of patients, mouse studies, extensive detailed mechanistic organoid analysis, scRNA-seq, and immunophenotyping. They observed that concentrations of this molecule predicted low mortality and low relapse risk in patients. In mice and in organoid models of intestinal stem cell health & T-cell-organoid co-culture experiments, that reduced GVHD in a STING-dependent and mTor-dependent fashion and modulates both gut epithelial function and the T cell compartment. Their model posits that DAT promotes proliferation & a regenerative phenotype of intestinal stem cells by activating mTOR signalling and STING activation. The effect of DAT could be recapitulated by inoculation mice with a bacterial strain that produces it

The study is a tour de force in its breadth of techniques and orthogonal approaches to evaluating the role of this metabolite. The figures are clear and the data compelling. The writing is precise, but the narrative of the results section would benefit greatly from the addition of interim concluding phrases and transitions. Taken together, these data show.... Having observed X, we next asked whether ... etc. The way it is written now without such interim concluding and transition phrases is rigorous/precise but dry/technical report of the data in the figures that is hard to read through given the size of the dataset

1C: disclose numbers of patients with an at-risk table below the survival curves. Do these associations hold up in multivariable analysis that considers key clinical factors? Do the concentrations of the metabolites predict outcomes well when they are considered as continuous variables and not just with binary splits into high vs low?

For all the organoid experiments when organoid counts and sizes are plotting, beginning with panel 2D and into the subsequent figures), in many instances the data are plotted as a ratio compared with control. That is fine as a visualization, but for a few key experiments please also plot the absolute values beside it so that the reader can be confident that the ratios are a good simplifying surrogate of underlying data.

LIne 258: the claim that metabolite treatment did not alter taxonomic composition is well-supported by the permanova test in Fig S3C, but the reader is first asked to visually inspect stacked barplots in Fig 3A and just agree visually with the assessment that the compositions are the same. Provide a statistical analysis, for example something very simple such as choosing the top handful of families and comparing their abundances with statistical test. Or, perhaps, measuring beta-diversity distances between individual mice and then comparing the distances between mice vs the distance between treatment groups.

Germ-free GVHD mouse experiments are not very common in the field and are an impressive logistical achievement. Please provide details on how this was done, particularly how mice were irradiated while maintaining them in a germ-free state

While most of the results are written very precisely to only claim what is supported by the data, the description of the transcriptomic results suffers from conflation of correlation and causality and should be revised. Line 382 claims metabolic activity of a cell type “was partly due to increased oxidative phosphorylation....” and then doubles down in line 384 “Consequently, we observed an upregulation...” casual claims cannot be made from the transcriptomic analysis, and oxphos was not directly (metabolically) measured. Please pull back description to talk about transcriptional programs/transcript abundance. Line 390 claims that mTORC1 signalling is “crucial” for the ISC to ISC-II transition is another causal claim. Line 397 has another casual claim “ISC-IIs required activation of mTORC1 signalling...”

The claim on line 399-400 that mTORC1 signalling and cellular stress responses were reduced was not measured directly. Only transcripts encoding machinery for these programs was observed to be reduced

Minor comments:

Fig 1B: consider plotting this with log-transformed y-axis to better visualize the distribution, especially since the median is later used as a relevant cutoff.

Line 128 replace the colloquial “level” with “concentration”

Fig 1J: confusing to plot cumulative incidence of relapse and TRM on the same graph; consider splitting these into separate panels

Line 183: explain (if I understood correctly) that these organoids were derived from Lgr5 reporter mice.

Line 254: “metabolite expression.” perhaps metabolite production or abundance?

Line 457: should Fig 6B actually say Fig 6C?

Reference # 18 is garbled (“T&#A0”)

(Remarks on code availability)

Version 1:

Reviewer comments:

Reviewer #1

(Remarks to the Author)

I appreciate the efforts of the authors to add new data to strengthen the manuscript. I have only minor concerns:

1. Fig 3C: Please explicitly specify the compared groups used for calculating the p-values. If compared with individual controls (i.e., FMT vehicle control, or GF vehicle control), the FMT ICA group and the GF ICA group should also have p-values. Similarly, please check Fig2C and 2D, Fig 6B and 6C.

2. L395-397: There appears to be a discrepancy between the presented data and the claimed conclusion. The authors claim that increased metabolic activity in DAT-treated organoids are more pronounced after irradiation. But the data show that after irradiation both control and DAT-treated groups had reduced oxygen consumption (i.e., more oxygen in the media, and cells have less OXPHOS activity). It is just that the DAT-treated group had a little less reduction of oxygen consumption compared with the control group. Please rephrase the sentence.

3. L448: “ICA-II cells” means “ISC-II cells”?

(Remarks on code availability)

Reviewer #2

(Remarks to the Author)

The authors did a great job in revising the manuscript. I have a minor suggestion on point 2 raised before.

The new data indicated that DAT administration strongly suppressed apoptosis, Lgr5 cell loss in WT but not in STING KO on day 1 and day 3. (new Fig. S5C-E). These findings should be clearly described and discussed. There is a wealth of information on blocking p53-dependent apoptosis and loss of Lgr5+ cells improving regeneration and survival of mice after TBI, using growth factors, Wnt agonist/GSK inhibitor and CDK4/6 inhibitors etc. Therefore, Lgr5+ ISC cell survival/preservation and proliferation (both) likely contribute to DAT-mediated intestinal protection. Yet, radiation-induced apoptosis and Lgr5 cell loss is independent of STING (ref 95, this ms). In my opinion, these data further strengthen the

critical role of epithelial STING in ISC regeneration.

(Remarks on code availability)

Reviewer #3

(Remarks to the Author)

this revision is responsive to my critiques.

(Remarks on code availability)

n/a

Point-to-point response to reviewer comments

A microbial metabolite protects against graft-versus-host disease via mTORC1 and STING-dependent intestinal regeneration

Götttert S. and Thiele-Orberg E. et al.

Reviewer #1

In the manuscript titled “A microbial metabolite protects against graft-versus-host disease via mTORC1 and STING-dependent intestinal regeneration” by Götttert et al., the authors have investigated the role of the microbial metabolite DAT in preventing GvHD in the context of allo-HSCT for leukemia therapy. The authors have demonstrated that DAT not only prevents GvHD but also promotes GvL responses. They have provided clear evidence that DAT promotes intestinal stem cell proliferation through mTORC1- and STING-dependent pathways, independent of type-I interferon receptor. However, the mechanism by which DAT modulates graft-versus-leukemia responses remains less clear. While this discovery is intriguing, there are some concerns that need further clarification.

Major Comments:

1) Did the number of crypt-like budding structures per spheroid also increase? If so, the role of DAT in intestinal stem cell (ISC) differentiation should also be investigated. Moreover, for consistency, the effect of DAT on ISC proliferation and irradiation-induced cell death should be examined *in vivo*.

As requested by the reviewer, we evaluated the number of budding structures per organoid. For consistency with the scRNA-Seq data, we did so on day 4 of metabolite treatment. As expected, based on our observations regarding organoid growth (**original Figure 2D**), an assay that assesses the number of viable crypt-like structures following mechanical disruption, we also observed increased organoid budding (**Reviewer’s Figure 1**). Regarding the role of DAT on ISC differentiation, please refer to the differential cell type abundance analysis performed by scRNA-Seq in the original manuscript (**original Figure 4H**).

Reviewer's Figure 1: Murine small intestinal organoids were stimulated with metabolites and the number of budding structures determined on day 4 of stimulation.

Regarding the effect of DAT on ISC abundance, proliferation, and cell death induction, please refer to the remarks of Reviewer #2 below.

2) Did the meropenem plus DAT treatment or *F. plautii* transfer alone actually improve GvHD and GvL in vivo? Additionally, does Sting-deficiency affect DAT's protective effect on GvHD and GvL in vivo?

A. The beneficial effect of DAT on GvHD in mice exposed to broad-spectrum antibiotics

We agree that it is important to determine whether exposure to the broad-spectrum antibiotic meropenem affects the efficacy of DAT treatment in protecting against GvHD in vivo. Our results show that DAT remains effective in promoting organoid recovery, a readout for damage to the intestinal stem cell compartment, even in dysbiotic, meropenem-treated mice (please refer to Figure 3E in the original manuscript). Similarly, our results show that administering the probiotic *F. plautii* to mice pre-treated with a broad-spectrum antibiotic combination (ampicillin/enrofloxacin) significantly improves organoid recovery. However, whether DAT can mitigate the detrimental effects of meropenem-induced dysbiosis and protect against GvHD is a highly relevant clinical scenario that may also have therapeutic implications. Therefore, we performed experimental GvHD in mice with meropenem-induced dysbiosis. As expected, meropenem treatment exacerbated GVHD severity. However, DAT treatment remained effective under dysbiotic conditions and significantly improved survival compared to mice receiving meropenem alone (**new Fig. 3F in the revised manuscript**).

The administration of the probiotic *F. plautii* was intended as proof of principle that DAT could be effective in promoting organoid recovery, whether supplied as a synthetic compound or naturally, via this DAT-producing probiotic. The dosing and timing of *F. plautii* treatment in GvHD require further optimization, a time-consuming process that is still in its early stages. Although we agree with Reviewer #1 that *F. plautii* could be a therapeutically promising probiotic that warrants further experimental investigation, an in-depth characterization of *F. plautii* is beyond the scope of the current manuscript. Given that Reviewer #3 has already highlighted the complexity and length of the manuscript, we have decided not to include these data in the current manuscript to ensure a focused and streamlined narrative.

B. The beneficial effect of DAT on GvL in mice exposed to broad-spectrum antibiotics

We agree with Reviewer #1 that whether exposure to broad-spectrum antibiotics compromises the beneficial effect of DAT treatment on GvL is an interesting question. Numerous clinical studies have highlighted a negative association between exposure to broad-spectrum antibiotics and the efficacy of immune-checkpoint inhibitors [1], CAR-T cell therapy [2, 3], and allogeneic stem cell transplantation [4, 5]. Since DAT's contribution to GvL responses in antibiotic-naive mice is only moderate (please refer to the **original Figures 1F-I**), exposing mice to meropenem will likely further reduce DAT's effect size. Given the principles of the 3Rs (Replacement, Reduction and Refinement) to which we are bound per regulatory statutes, it may not be feasible to use large numbers of mice to achieve the necessary group sizes for statistical power to detect DAT's beneficial effect in antibiotic-treated mice, given that this question extends the current scope of the manuscript. In light of Reviewer #3's comments regarding the current breadth of the manuscript, we suggest omitting the meropenem-GvL data in this revision.

However, Reviewer #1 raises an essential point regarding DAT's ability to maintain or promote GvL. Therefore, we included an additional clinically relevant GvL model targeting a different cancer type to highlight further that DAT contributes to anti-leukemia responses. We observed that DAT treatment significantly reduced relapse rates in the FLT3-ITD/MLL-PTD AML model compared to treatment with allogeneic T cells alone without aggravating GvHD (**new Fig. S1D-E in the revised manuscript**).

C. STING-deficiency on GvHD and GvL

The current manuscript shows that DAT treatment requires intact STING signaling for its beneficial effects, both *in vitro* in organoids and *in vivo* in the allo-BMT mouse model, as shown via our organoid regeneration assay (**Fig. 5 of the original manuscript**). In previous publications, we and others addressed the role of STING in GvHD (e.g., Fischer et al. [6]): STING-deficient mice have a reduced capacity for intestinal regeneration, and GvHD is exacerbated in these mice. GvL responses in STING-deficient mice remain largely unexplored, and a thorough investigation of this would exceed the current scope of our manuscript. Given that we demonstrate in **Figures 4A-D** that DAT treatment is inefficient in STING-deficient mice, and in the interest of adhering to the principles of the 3Rs, we suggest omitting GvHD/GvL experiments with DAT treatment in STING-deficient mice in this revision.

However, Reviewer #1 raises an essential point regarding whether STING is required in the epithelium (as opposed to hematopoietic cells) for DAT's protective effects. Therefore, we employed an additional, conditional genetic knock-out model, using STING^{fl/fl} x Villin-Cre mice, to investigate whether intestinal epithelial-specific gene deletion of STING abrogates DAT's effect. We found that DAT's beneficial effect on intestinal regeneration is abrogated in mice lacking epithelial-intrinsic STING (**new Fig. S4C in the revised manuscript**).

3) The effects of DAT on Tregs and CD8 T cells appear inconsistent between the *in vitro* and *in vivo* data, especially at low doses of DAT. Is it possible that the promoting effect of DAT on GvL is through innate immune activation, considering that DAT administration was stopped before allo-T cell transfer?

We agree that our *in vivo* results partly differ from our *in vitro* observation following stimulation of purified human T cells (**Fig. 6 of the original manuscript**). To address Reviewer #1's question about whether DAT could affect innate immune cells, we evaluated the dynamics, antigen presentation via MHC-II expression, and innate activation via costimulatory molecules (CD80 and CD86) in dendritic cells and monocytes isolated from the ileal lamina propria and mesenteric lymph nodes (MLN). As shown in the **Reviewer's Figure 2**, we observed a decrease in the number of myeloid cells infiltrating mesenteric lymph nodes after treatment with DAT and ICA. ICA treatment further reduced the expression of MHC-II and CD86 on macrophages and monocytes in MLN and of MHC-II on monocytes in the lamina propria. These findings are consistent with our observations in intestinal organoids and human monocytes (**Fig. S2B and S6K of the original manuscript**). In contrast, DAT treatment did not affect the antigen presentation or activation of myeloid cells.

Reviewer's Figure 2: C57BL/6 mice underwent allo-BMT, and myeloid cell infiltration into the mesenteric lymph nodes was assessed. Expression levels of MHC class II and CD86 on myeloid cells from the mesenteric lymph nodes and lamina propria were analyzed by flow cytometry.

Minor Comments:

1) Why was DAT administered only for 7 days before allo-BMT (Fig. 1D) or 9 days before allo-T cell transfer (Fig. 1F), while ICA was given continuously before and after allo-BM and allo-T cell transfer? If DAT is given at a relatively low dose continuously, would its protective effect disappear? How rapidly do serum levels of DAT drop after discontinuing administration?

When establishing the model, we evaluated different DAT treatment regimens and used the treatment regimen for ICA published by Swimm et al. [7] as a reference. Among the tested DAT treatments, the one shown in the original manuscript performed the best. When administered in lower doses continuously (from day -7 onwards) ad libitum via drinking water, DAT treatment yields only a moderate survival benefit, as shown in **Reviewer's Figure 3**.

Reviewer's Figure 3: Survival of mice. Balb/c mice were treated with low-dose DAT via the drinking water from d-7 onwards throughout the experiment and underwent allo-BMT.

To address Reviewer #1's question regarding serum levels, we analyzed serum concentrations of DAT following oral administration in mice (**Reviewer's Figure 4**). Generally, DAT was not detected in untreated mice. Following oral administration, serum DAT levels increased and reached concentrations comparable to those used in *in vitro* experiments within 20 minutes. Although serum levels declined substantially by 3 hours post-administration, DAT remained detectable..

Reviewer's Figure 4: DAT serum concentrations 20 minutes or 3h after oral application.

2) Typo errors: For example, in Fig. 1C and I, “probability of survival” should be “percentage of survival”; in Fig. 1J, “probability” should be “relapse rate”; in the main text L457, “Fig 6B” should be “Fig 6C”.

We thank the reviewer for the comment and have corrected the errors accordingly.

Reviewer # 2

This is an interesting study to show that the levels of microbial product desaminotyrosine (DAT) are associated with better outcome in allo-HSCT patients. Pretreatment of mice with DAT or DAT producing probiotics improved survival and intestinal regeneration of allo-HSCT after total body irradiation (TBI). The benefit of DAT was abrogated in germ-free or STING-deficient (KO, Gt/Gt) mice. The GI protective effects in vivo (survival, histology, inflammation and Lgr5 preservation) are supported by strong data. In organoids, DAT promotes growth of untreated culture as well and that under many conditions, i.e, activated T cells, radiation, chemo, with or without EGF, germ-free or antibiotic treatment. STING KO but not IFNAR1 KO organoids showed loss of DAT induced growth. The author previously published that DAT promotes T cell-dependent anticancer immunity. In this study, the contribution from T cell activation or immune modulation seems minor but perhaps not surprising as the compound was given before TBI and allograft. Overall, the approach is comprehensive, and the current data support a cell intrinsic protective mechanism. However, the mechanistic studies missed a critical aspect on injury protection/reduction as detailed below.

Major Comments:

1) DAT was given for 7 days before TBI and allograft. Even high-level DAT pretreatment was trending protection (Fig. 2 and 7C). Is continuous treatment of low dose also detrimental?

Please refer to our Response to Reviewer #1's minor comment 1) and **Reviewer's Figure 3**. Continuous treatment with low-dose DAT administered ad libitum via drinking water is not as effective as oral gavage in mitigating GvHD and prolonging survival, but it is not detrimental to survival.

2) Fig. 5, the growth promoting phenotype in organoids needs confirmation in mice by accessing proliferation, mTOR/metabolic signaling, and Lgr5 cell expansion in WT and STING KO mice after 7 day of DAT treatment, but before TBI and allograft (day 0). Day 7 is outcome.

Fig. S2C, 2G indicated that DAT pretreatment resulted in radio and chemo resistance in organoids, and the barrier effects in mice were pronounced on day 3. TBI causes acute crypt damage through p53-PUMA dependent loss of Lgr5⁺ stem and TA cells, which is exacerbated likely by allograft T cells. Activation of the PI3K/AKT and Wnt pathways (growth factors, Rspodin, GSK inhibitors) blunts Lgr5 and crypt apoptosis and improves intestine regeneration, only when given prior to exposure. Fig. 5G. does DAT treatment affect Lgr5⁺ cell numbers, crypt apoptosis/death (TUNEL) on 0, 1 and 3, and any effects on the p53 pathway or targets?

As suggested by Reviewers #1 and #2, we assessed the number of Lgr5⁺ intestinal stem cells (ISCs), epithelial proliferation by quantifying the number of Ki67⁺ cells per intestinal crypts, and apoptosis by cleaved caspase-3 staining (1) at steady state after 7 days of DAT treatment and (2) at days 1 and 3 following total body irradiation in WT and STING KO mice.

At steady state, and the absence of epithelial injury, DAT treatment did not induce an expansion of intestinal stem cells or proliferating cells. However, at days 1 and 3 following damage by total body irradiation, DAT treatment increased the number of Ki67⁺ proliferating cells and Lgr5⁺

intestinal stem cells compared to control-treated mice. Consistent with this regenerative effect, cleaved caspase-3⁺ apoptotic cell numbers were significantly reduced in DAT-treated mice compared to controls.

In contrast, in STING^{GT/GT} mice, DAT treatment did not affect the number of Lgr5⁺ intestinal stem cells, nor did it impact apoptosis following TBI. However, the number of Ki67⁺ proliferating crypt cells at days 1 and 3 after TBI remained partly elevated following DAT treatment, independent of STING (**new Fig. S5C-E**).

These findings are consistent with our observations of our scRNA-Seq, indicating that STING is required to maintain proliferation and an intestinal stem cell state following DAT stimulation (**Fig. 4E-H of the original manuscript**).

3) Fig. 4 In organoids, STING-dependent growth promoting phenotype is novel and interesting, but cannot be directly extrapolated to the in vivo injury model. mTORi or antioxidant is not STING specific. It is difficult to understand “deregulated mTOR signaling” or metabolism by STING KO based on pathway analysis alone. Inclusion of relevant DEG tables can be helpful. Radiation should be used to better address cell intrinsic mechanism. Does STING KO abrogated DAT-mediated radioprotection, how about the effects on mTOR signaling/metabolism, apoptosis, and the p53 pathway?

As requested by the Reviewer, we now supply DEG tables with this revised submission.

In the revised manuscript, we now assess organoid death and regeneration following irradiation in STING^{GT/GT} organoids. Following irradiation with 4 Gy, we observed that DAT treatment improved organoid growth in WT compared to control-treated organoids, but had no effect in STING^{GT/GT} organoids (**new Fig. S4B**).

We further investigated the oxygen consumption of organoids as a surrogate for metabolic activity at steady state or following irradiation. DAT treatment increased oxygen consumption in WT organoids compared to control-treated organoids (**new Fig. S4F**). As expected, irradiation markedly reduced oxygen consumption. In WT organoids, DAT treatment maintained the metabolic activity of organoids compared to control organoids. This effect was abrogated in STING KO organoids (**new Fig. S4G**).

4) Fig 8. The immune compartment and the ISC compartment seem disconnected in “injury protection” model.

We thank the reviewer for this comment. We have now updated Fig. 8 accordingly.

5) Discussion. “Type 1 IFN-independent”. Prior studies established that epithelial intrinsic STING and IFN beta is required for intestinal regeneration after TBI (PMID: 28424327; PMID: 34613772). Either loss of DAT-induced Lgr5 expansion before injury or failed regeneration after can lead to similar outcomes in STING KO mice. This point requires discussion.

We thank the reviewer for this comment. The additional data provided with this revision further support (1) the requirement of epithelial-intrinsic STING, demonstrated by our new experiments in STING^{fl/fl} x Villin-cre mice (please see our reply to Reviewer #1's comment 2C), and (2) now clarify the role of STING in mediating the effects of DAT at steady state vs. early after TBI as requested by Reviewer #2. We have also updated the discussion accordingly and added the suggested references.

Reviewer #3

Gottert & coauthors studied the role of desaminotyrosine (DAT), a microbiome-derived metabolite, in GVHD. This comprehensive study made use of observational cohorts of patients, mouse studies, extensive detailed mechanistic organoid analysis, scRNA-seq, and immunophenotyping. They observed that concentrations of this molecule predicted low mortality and low relapse risk in patients. In mice and in organoid models of intestinal stem cell health & T-cell-organoid co-culture experiments, that reduced GVHD in a STING-dependent and mTor-dependent fashion and modulates both gut epithelial function and the T cell compartment. Their model posits that DAT promotes proliferation & a regenerative phenotype of intestinal stem cells by activating mTOR signalling and STING activation. The effect of DAT could be recapitulated by inoculation mice with a bacterial strain that produces it

The study is a tour de force in its breadth of techniques and orthogonal approaches to evaluating the role of this metabolite. The figures are clear and the data compelling. The writing is precise, but the narrative of the results section would benefit greatly from the addition of interim concluding phrases and transitions. Taken together, these data show.... Having observed X, we next asked whether ... etc. The way it is written now without such interim concluding and transition phrases is rigorous/precise but dry/technical report of the data in the figures that is hard to read through given the size of the dataset

We thank the reviewer for that kind evaluation of our manuscript. We incorporated additional interim conclusions and transitions, as suggested.

Major comments

1) 1C: disclose numbers of patients with an at-risk table below the survival curves. Do these associations hold up in multivariable analysis that considers key clinical factors? Do the concentrations of the metabolites predict outcomes well when they are considered as continuous variables and not just with binary splits into high vs low?

We have updated Figure 1C and 1J to include the corresponding at-risk tables.

Regarding the key clinical factors: In a previous study [5], we described the immunomodulatory metabolic risk index (IMM-RI), which comprises the two metabolites evaluated here, as well as the short-chain fatty acids propionic and butyric acid, and the branched-chain fatty acid isovaleric acid. In that cohort, IMM-RI stratification into low- and high-risk showed no significant association with sex, HCT-CI, disease risk, conditioning intensity, or antibiotic exposure. In the present study, the sample size (n = 50 patients for calculating the cut-offs for DAT and ICA via the Youden Index) was insufficient for a robust multivariable analysis; thus, results should be interpreted as exploratory and hypothesis-generating.

As requested, we also analyzed DAT and ICA concentrations as continuous variables using Cox regression. As a continuous variable, neither metabolite was significantly associated with overall survival (DAT: p = 0.123, HR = 0.000, 95% CI: 0.000–26.8; ICA: p = 0.119, HR = 0.000, 95% CI: 0.000–478.3).

2) For all the organoid experiments when organoid counts and sizes are plotting, beginning with panel 2D and into the subsequent figures), in many instances the data are plotted as a ratio compared with control. That is fine as a visualization, but for a few key experiments please also plot the absolute values beside it so that the reader can be confident that the ratios are a good simplifying surrogate of underlying data.

We agree with the reviewer that plotting data as ratios is a simplification. We intentionally used ratio-based plots to highlight treatment effects across different organoid lines or over time, as absolute values can vary considerably between experiments due to inherent differences in organoid growth behavior. To address this variability and still allow for comparison, we presented unnormalized organoid size data in the original manuscript (e.g., in Fig. 2 B of the original manuscript).

To satisfy this important consideration, we now offer the requested plots of raw Lgr5⁺ ISCs (corresponding to Fig. 2C) and organoid counts per well (corresponding to Fig. 2D) in **Reviewer's Figure 5**.

Reviewer's Figure 5: Raw data plots of the original figures 2C and 2D. Left: abundance of intestinal stem cells determined by flow cytometry of small intestinal organoids derived from Lgr5-GFP reporter mice as percentage of Lgr5-GFP^{high} cells of EpCAM⁺ cells (referring to 2C). Right: number of organoids counted per individual well after passaging (referring to 2D).

3) Line 258: the claim that metabolite treatment did not alter taxonomic composition is well-supported by the permanova test in Fig S3C, but the reader is first asked to visually inspect stacked barplots in Fig 3A and just agree visually with the assessment that the compositions are the same. Provide a statistical analysis, for example something very simple such as choosing the top handful of families and comparing their abundances with statistical test. Or, perhaps, measuring beta-diversity distances between individual mice and then comparing the distances between mice vs the distance between treatment groups.

As suggested by the reviewer, we now show the relative abundances of the top 5 families (Erysipelotrichaceae, Lachnospiraceae, Lactobacillaceae, Muribaculaceae and Prevotellaceae) and the corresponding statistical tests (ordinary one-way ANOVA with Dunnett's correction for multiple comparisons) in the **revised Fig. 3A**. As expected, we observed no statistically significant difference in abundance of these families between DAT and control-treated mice, with the exception of Prevotellaceae, which is less abundant in DAT-

treated mice compared to controls.

4) Germ-free GVHD mouse experiments are not very common in the field and are an impressive logistical achievement. Please provide details on how this was done, particularly how mice were irradiated while maintaining them in a germ-free state

We thank the reviewer for that kind feedback. We added additional details to the method section to clarify how irradiation was performed in GF mice.

5) While most of the results are written very precisely to only claim what is supported by the data, the description of the transcriptomic results suffers from conflation of correlation and causality and should be revised. Line 382 claims metabolic activity of a cell type “was partly due to increased oxidative phosphorylation...” and then doubles down in line 384 “Consequently, we observed an upregulation...” casual claims cannot be made from the transcriptomic analysis, and oxphos was not directly (metabolically) measured. Please pull back description to talk about transcriptional programs/transcript abundance. Line 390 claims that mTORC1 signalling is “crucial” for the ISC to ISC-II transition is another causal claim. Line 397 has another casual claim “ISC-IIs required activation of mTORC1 signalling...”

The claim on line 399-400 that mTORC1 signalling and cellular stress responses were reduced was not measured directly. Only transcripts encoding machinery for these programs was observed to be reduced

We thank the reviewer for this feedback. We revised the description of our transcriptomic results to recapitulate the observations more precisely. We additionally added data supporting metabolic activation of organoids in the new **Fig. S4F** as described in response to Reviewer #2's major comment 3.

Minor comments:

1) Fig 1B: consider plotting this with log-transformed y-axis to better visualize the distribution, especially since the median is later used as a relevant cutoff.

As suggested by the reviewer, we visualized the data of Fig. 1B with a log-transformed y-axis in **Reviewer's Figure 6**. However, for the manuscript, we prefer to stick to the original visualisation.

Reviewer's Figure 6: Metabolite data of the original Figure 1 B visualized with log-transformed y-axis.

2) Line 128 replace the colloquial "level" with "concentration"

We changed „levels“ to „concentrations“ throughout the text.

3) Fig IJ: confusing to plot cumulative incidence of relapse and TRM on the same graph; consider splitting these into separate panels

Due to spacial constraints (we now show the numbers at risk in the revised **Figure 1C and J**), we continue to show cumulative incidence on the same graph, which is not uncommon in the reporting of competing risks between TRM/Relapse and GVHD/Death and is a standard output of the R package used for this analysis.

4) Line 183: explain (if I understood correctly) that these organoids were derived from Lgr5 reporter mice.

We clarified that aspect in the corresponding figure legend.

5) Line 254: "metabolite expression." perhaps metabolite production or abundance?

We changed it to metabolite production as suggested.

6) Line 457: should Fig 6B actually say Fig 6C?

Yes, we corrected that part accordingly.

7) Reference # 18 is garbled ("T&#A0")

We corrected the mentioned reference.

References:

1. Routy, B., et al., *Gut microbiome influences efficacy of PD-1–based immunotherapy against epithelial tumors*. *Science*, 2018. **359**(6371): p. 91-97.
2. Smith, M., et al., *Gut microbiome correlates of response and toxicity following anti-CD19 CAR T cell therapy*. *Nat Med*, 2022. **28**(4): p. 713-723.
3. Stein-Thoeringer, C.K., et al., *A non-antibiotic-disrupted gut microbiome is associated with clinical responses to CD19-CAR-T cell cancer immunotherapy*. *Nat Med*, 2023. **29**(4): p. 906-916.
4. Shono, Y., et al., *Increased GVHD-related mortality with broad-spectrum antibiotic use after allogeneic hematopoietic stem cell transplantation in human patients and mice*. *Science Translational Medicine*, 2016. **8**(339): p. 339ra71-339ra71.
5. Thiele Orberg, E., et al., *Bacteria and bacteriophage consortia are associated with protective intestinal metabolites in patients receiving stem cell transplantation*. *Nature Cancer*, 2024.
6. Fischer, J.C., et al., *RIG-I/MAVS and STING signaling promote gut integrity during irradiation- and immune-mediated tissue injury*. *Sci Transl Med*, 2017. **9**(386).
7. Swimm, A., et al., *Indoles derived from intestinal microbiota act via type I interferon signaling to limit graft-versus-host disease*. *Blood*, 2018. **132**(23): p. 2506-2519.

Point-to-point response to reviewer comments

A microbial metabolite protects against graft-versus-host disease via mTORC1 and STING-dependent intestinal regeneration

Göttert S. and Thiele-Orberg E. et al.

Reviewer #1

I appreciate the efforts of the authors to add new data to strengthen the manuscript. I have only minor concerns:

1. Fig 3C: Please explicitly specify the compared groups used for calculating the p-values. If compared with individual controls (i.e., FMT vehicle control, or GF vehicle control), the FMT ICA group and the GF ICA group should also have p-values. Similarly, please check Fig 2C and 2D, Fig 6B and 6C.

In Fig. 3C, the experimental groups were compared with their respective individual controls, e.g. DAT + FMT vs vehicle-control + FMT (left), or DAT in GF vs vehicle-control in GF (right). Throughout the manuscript, we occasionally omitted p-values when no significant differences between group means were observed, in order to improve readability. However, as requested, we have now included the additional statistical information in the relevant figures.

2. L395-397: There appears to be a discrepancy between the presented data and the claimed conclusion. The authors claim that increased metabolic activity in DAT-treated organoids are more pronounced after irradiation. But the data show that after irradiation both control and DAT-treated groups had reduced oxygen consumption (i.e., more oxygen in the media, and cells have less OXPHOS activity). It is just that the DAT-treated group had a little less reduction of oxygen consumption compared with the control group. Please rephrase the sentence.

We thank the reviewer for that comment. We rephrased the sentence to precisely recapitulate the finding.

3. L448: "ICA-II cells" means "ISC-II cells"?

That is correct. We corrected the error accordingly.

Reviewer #2

The authors did a great job in revising the manuscript. I have a minor suggestion on point 2 raised before. The new data indicated that DAT administration strongly suppressed apoptosis, Lgr5 cell loss in WT but not in STING KO on day 1 and day 3. (new Fig. S5C-E). These findings should be clearly described and discussed. There is a wealth of information on blocking p53-dependent apoptosis and loss of Lgr5+ cells improving regeneration and survival of mice after TBI, using growth factors, Wnt agonist/GSK inhibitor and CDK4/6 inhibitors etc. Therefore, Lgr5+ ISC cell survival/preservation and proliferation (both) likely contribute to DAT-mediated intestinal protection. Yet, radiation-induced apoptosis and Lgr5 cell loss is independent of STING (ref 95, this ms). In my opinion, these data further strengthen the critical role of epithelial STING in ISC regeneration.

We thank the reviewer for the kind feedback. We have expanded both the Results and Discussion sections to highlight the new data and address the points raised.

Reviewer #3

this revision is responsive to my critiques.

We thank the reviewer for that feedback.